# Neural Spectral Methods:
## Self-supervised learning in the spectral domain

**Yiheng Du, Nithin Chalapathi, Aditi S. Krishnapriyan**
`{yihengdu, nithinc, aditik1}@berkeley.edu`

University of California, Berkeley

## Abstract

We present Neural Spectral Methods, a technique to solve parametric Partial Differential Equations (PDEs), grounded in classical spectral methods. Our method uses orthogonal bases to learn PDE solutions as mappings between spectral coefficients, instantiating a spectral-based neural operator. In contrast to current machine learning approaches which enforce PDE constraints by minimizing the numerical quadrature of the residuals in the spatiotemporal domain, we leverage Parseval's identity and introduce a new training strategy through a *spectral loss*. Our spectral loss enables more efficient differentiation through the neural network, and substantially reduces training complexity. At inference time, the computational cost of our method remains constant, regardless of the spatiotemporal resolution of the domain. Our experimental results demonstrate that our method significantly outperforms previous machine learning approaches in terms of speed and accuracy by one to two orders of magnitude on multiple different problems, including reaction-diffusion, and forced and unforced Navier-Stokes equations. When compared to numerical solvers of the same accuracy, our method demonstrates a $10\times$ increase in performance speed. Our source code is publicly available at `https://github.com/ASK-Berkeley/Neural-Spectral-Methods`.

## 1 Introduction

Partial differential equations (PDEs) are fundamental for describing complex systems like turbulent flow (Temam, 2001), diffusive processes (Friedman, 2008), and thermodynamics (Van Kampen, 1992). Due to their complexity, these systems frequently lack closed-form analytical solutions, prompting the use of numerical methods. These numerical techniques discretize the spatiotemporal domain of interest and solve a set of discrete equations to approximate the system's behavior.

Spectral methods are one such class of numerical techniques, and are widely recognized for their effectiveness (Boyd, 2001; Gottlieb & Orszag, 1977). These methods approximate PDE solutions as a sum of basis functions and transform the equations into the spectral domain. Spectral methods are known for their fast convergence and computational efficiency, especially for problems with smooth solutions. They are notably impactful in fields like computational fluid dynamics (Peyret, 2002).

Numerical methods can be computationally expensive because a fine discretization of the physical domain and a large number of time-stepping iterations are often required to achieve high accuracy. Additionally, many engineering applications require solving systems under different parameters or initial conditions, necessitating multiple iterations to repeatedly solve such systems. The aforementioned spectral methods also rely on time-stepping schemes, and present similar challenges to other numerical methods. These factors underscore the need for more efficient computational strategies.

Recent advances in machine learning (ML) highlight neural networks (NNs) as potential alternatives or enhancements to traditional numerical solvers. Consider PDEs on a regular domain $\Omega \subseteq \mathbb{R}^d$:

$$\begin{aligned}
\mathcal{F}_\phi(u(x)) &= 0 \quad \text{in } \Omega, \\
\mathcal{B}_\phi(u(x)) &= 0 \quad \text{on } \partial\Omega,
\end{aligned} \tag{1}$$

where $u$ is the classical solution, $\mathcal{F}_\phi$ is the potentially nonlinear differential operator, and $\mathcal{B}_\phi$ are the boundary condition(s). Both operators are parameterized by $\phi$, which could correspond to initial conditions or parameters associated with $\Omega$, such as diffusion coefficients. A common ML approach

is to train NNs on datasets of numerical solutions spanning various parameters, and then learn the mapping $\mathcal{G}_\theta : \phi \mapsto u_\theta$ from parameters to solutions. This approach is typically done via supervised learning, where the NN is trained by minimizing the error between the predicted solution and the numerical solution. At inference time, the goal is to generalize to previously unseen parameters and directly predict solutions. By amortizing computational costs during training and having rapid inference, these data-driven approaches have the potential to be more efficient than traditional numerical solvers.

When there is information about the governing physical laws, another common ML approach is to train the NN through a loss function that imposes a soft constraint on these laws (Raissi et al., 2019). In this case, the NN predicts a solution $u_\theta$, and then the corresponding residual function, $R(x) = \mathcal{F}_\phi(u_\theta(x))$, is minimized on a set of sampled spatial and/or temporal points. Specifically, an additional loss term is defined as the numerical quadrature to the residual norm:

$$L_{\mathsf{PINN}}(u_\theta) := \frac{1}{N} \sum_{n \in [N]} R(x_n)^2 \approx ||R(x)||^2_{\mathcal{L}^2(\Omega)}, \quad x_n \sim \text{i.i.d. } \mathcal{U}(\Omega). \tag{2}$$

This is often called a Physics-Informed Neural Network (PINN) loss function, which we denote as $L_{\mathsf{PINN}}(u_\theta)$. In practice, this approach does not require solution data on the interior of the domain, and the NN can be trained by only satisfying the PDE(s) of interest. This loss function can also be combined with a data fitting loss and trained together. However, this can often be impractical, as it requires knowing both the underlying governing physical laws *and* having solution data (either through a numerical solver or through observational measurements).

Most current ML methods to solve PDEs (Kochkov et al., 2021; Li et al., 2020) are grid-based approaches. To model parametric solutions, these models create a mesh, and perform the parameters-to-solutions mapping through non-local transformations on function values at points in the mesh. One such example are neural operators (Kovachki et al., 2021), which parameterize the mapping using iterative kernel integrals, and have been applied to a wide range of engineering problems (Zhang et al., 2022; Kurth et al., 2023). Neural operators can be trained through supervised learning procedures (a loss function that matches solution data), self-supervised methods (such as the aforementioned PINN loss function), and a combination of both.

Data-fitting and PINN loss approaches both have several limitations:

- **Data availability.** The effectiveness of data-driven methods generally depends on the availability of large datasets consisting of PDE parameters and corresponding solutions. For complex problems, solution data can only be generated from expensive solvers (or through observations), and also includes inherent numerical errors. When solving new systems, new solution data often needs to be generated, which can be time-consuming.

- **Optimization.** Empirical evidence has suggested that minimizing the PINN loss often encounters convergence issues, particularly for complex problems, resulting in subpar accuracy. This is likely attributed to the ill-posed nature of the optimization problem that arises from incorporating PDE constraints into the model (Krishnapriyan et al., 2021; Wang et al., 2021a).

- **Computation cost.** Computing the PINN loss involves evaluating the differential operator $\mathcal{F}$ at sampled points, which requires computing higher-order derivatives of $u_\theta$. As the complexity of $u_\theta$ increases, the computation cost of back-propagation scales significantly. Moreover, accurate estimation of the residual norm requires a substantial amount of sampled points to enforce the PDE. For neural operators such as the commonly used Fourier Neural Operator (FNO) Li et al. (2020), differentiation costs scale quadratically with the number of points. This is due to the use of Fourier transform, and this makes it intractable to take exact derivatives for large numbers of sampled points (see discussions in §B).

To address these issues with accuracy and efficiency, our work explores the incorporation of ML with spectral methods. We focus on a data-constrained setting to learn the solution operator of parameterized PDEs, where we assume that we have no solution data on the interior of the spatiotemporal domain and only train our model by minimizing the PDE residual. Given the form of a differential operator $\mathcal{F}_\phi$, the model learns to map the parameter function $\phi$ to the corresponding solution $u_\theta$. Our key insights are to learn the solution as a series of orthogonal basis functions, and leverage Parseval's Identity to obtain a loss function in the spectral domain. While prior approaches

minimize the approximated residual function by computing higher-order derivatives on the sampled points, our method exploits properties of the spectral representation to obtain the exact residual via algebraic operations on the prediction.

Our contributions are summarized as follows:

- We propose Neural Spectral Methods (NSM) to learn PDE solutions in the spectral domain. Our model parameterizes spectral transformations with NNs, and learns by minimizing the norm of the residual function in the spectral domain. Since solution data can be expensive to generate for every new problem, we focus on scenarios with no solution data on the interior of the domain.

- We introduce a spectral-based neural operator that can learn transformations in a spectral basis. By operating on fixed collocation points, our proposed spectral-based neural operator avoids aliasing error and avoids scaling the computational cost with grid resolution.

- We introduce a spectral loss to train models entirely in the spectral domain. By utilizing the spectral representation of the prediction and Parseval's identity, the residual norm is computed by exact operations on the spectral coefficients. This approach avoids sampling a large number of points and the numerical quadrature used by the PINN loss, thereby simplifying computation complexity.

- We provide experimental results on three PDEs: Poisson equation §4.1, Reaction-Diffusion equation §4.2, and Navier-Stokes equations §4.3. Our approach consistently achieves a minimum speedup of $100\times$ during training and $500\times$ during inference. It also surpasses the accuracy of grid-based approaches trained with the PINN loss by over $10\times$. When tested on different grid resolutions, our method maintains constant computational cost and solution error. In comparison to iterative numerical solvers that achieve equivalent accuracy, our method is an order of magnitude faster.

## 2 RELATED WORKS

**ML methods for solving PDEs.** Using NNs to solve PDEs has become an active research focus in scientific computing (Han et al., 2018; Raissi et al., 2019; Lu et al., 2021). He et al. (2018); Mitusch et al. (2021) explore finite element methods in NNs. Yu et al. (2018); Sirignano & Spiliopoulos (2018); Ainsworth & Dong (2021); Bruna et al. (2022) recast PDEs into variational forms and apply the Galerkin projection minimization via sampling-based losses. Sharma & Shankar (2022) accelerate discretized PDE residual computation. Wang et al. (2021b) focus on Fourier features and eigenvalue problems. In the context of spectral methods, Lange et al. (2021) focus on data-fitting; Dresdner et al. (2022) learn to correct numerical solvers; Xia et al. (2023); Lütjens et al. (2021) use spectral representations in spatial domains; and Meuris et al. (2023) extract basis functions from trained NNs for downstream tasks. These studies differ in problem settings and deviate from our method in both architecture and training.

**Neural operators.** Neural operators (Kovachki et al., 2021; Li et al., 2020; Gupta et al., 2021) learn mappings between functions, such as PDE parameters or initial conditions to the PDE solutions. They are typically trained in a supervised learning manner on parameter–solution datasets, and aim to learn the functional mappings between them (i.e., the solution operator). However, the supervised learning setting poses a challenge in data generation for new or complex problems, especially when the data is scarce or the numerical solvers generating it are inefficient.

One of the most common neural operators is the Fourier Neural Operator (FNO) (Li et al., 2020; Kurth et al., 2023; Zhang et al., 2022). The training process for FNO consists of performing convolutions through Fourier layers and learning in the frequency domain. The Spectral Neural Operator (SNO) (Fanaskov & Oseledets, 2022) was proposed to reduce aliasing error in general neural operators by utilizing a feed-forward network to map between spectral coefficients. Similarly, the TransformOnce (T1) (Poli et al., 2022) model looks at learning transformations in the frequency domain with an improved model architecture. However, both models have a number of architecture and training differences from NSM, and as we will show, have much poorer accuracy. They also only look at the supervised learning setting.

**Physics-Informed Neural Networks (PINNs).** The physics-informed neural networks (PINNs) framework (Raissi et al., 2019) adds the governing physical laws (i.e., PDE residual function), estimated on sampled points, into the NN's loss function. This additional term, which we refer to as a

*PINN loss* (Eq. 2), acts as a soft constraint to regularize the model's prediction of the PDE solution, and can be considered a self-supervised loss function. This approach can also be used in a operator learning setting across various architectures, where the base architecture is a neural operator and the PINN loss is used to train the model Li et al. (2021); Tripura et al. (2023); Rosofsky et al. (2023); Goswami et al. (2022).

However, the PINN approach requires evaluating the PDE residual on a large number of sampled points in the interior domain. In scenarios with higher-order derivatives in the PDE residual, multiple differentiations through the NN are required. When using the PINN loss with grid-based neural operators such as FNO, the Fast Fourier Transform (FFT) in the forward pass escalates to quadratic in batched differentiation with respect to the number of sampled points, making exact residual computation through auto-differentiation computationally expensive (see discussions in §B for more details). As we will show, these *grid-based* methods are inaccurate and often overfit to the training grid size because of aliasing error.

## 3 NEURAL SPECTRAL METHODS

We introduce Neural Spectral Methods (NSM), a general class of techniques to incorporate spectral methods with NNs. NSM learns the mapping from PDE parameters to PDE solutions, i.e., $\mathcal{G}_\theta : \phi \mapsto u_\theta$, and is shown in Fig. 1. NSM consists of two key components: a base NN architecture (Fig. 1a) that maps the spectral representation of the parameters $\phi$ to that of its solutions $u_\theta$, and a spectral training procedure (Fig. 1b) that minimizes the spectral norm of the PDE residual function.

In §3.2, we describe our core NN architecture, which incorporates spectral methods within a neural operator framework. In §3.3, we introduce our spectral training procedure. Finally, in §4, we demonstrate the strong empirical performance of NSM for learning solutions to different PDE classes.

### 3.1 BACKGROUND

**Notation.** A set of functions on $\Omega$ is denoted as $\{f_m(x) : x \in \Omega\}_{m \in \mathcal{I}}$ where $\mathcal{I}$ is a countable index set. We denote integer indices by $[n] := \{1, 2, .., n\}$. The $n$th component of a vector $x$ is denoted as $x_n$. Given a set of basis functions, the spectral coefficients of a function $u$ are denoted as $\tilde{u}$. For an integer $k > 0$, the $k$th order Sobolev space (Evans, 2022) on domain $\Omega \subseteq \mathbb{R}^d$ is denoted as $\mathcal{H}^k(\Omega)$.

**Orthogonal basis.** Orthogonal basis functions are fundamental components of spectral methods. For completeness, the definition of orthogonality is provided in §A. The choice of the basis functions is problem-specific and they must have desirable spectral properties. In this paper, we focus on two commonly used orthogonal bases for periodic and other types of boundary conditions, respectively:

**Example 1** (Fourier basis). $\{\sin(mx), \cos(mx) : x \in 2\pi\mathbb{T}\}_{m \in \mathbb{N}}$ *w.r.t Lebesgue measure.*

**Example 2** (Chebyshev polynomials). $\{T_m(x) : x \in [-1, 1]\}_{m \in \mathbb{N}}$ *w.r.t* $\frac{d\mu}{dx} = 1/\sqrt{1-x^2}$, *where*

$$T_m(x) := \cos(m \cos^{-1}(x)) \equiv (-1)^m \sin(m \sin^{-1}(x)).$$

For multi-dimensional problems, we can verify that the product of basis in each dimension preserves completeness and orthogonality (see Proposition 1). Spectral representations are known for their efficiency in representing smooth functions. Specifically, Fourier and Chebyshev interpolations possess the well-known spectral decay for sufficiently smooth functions (Mason & Handscomb, 2002):

**Fact 1.** *For any* $f \in \mathcal{H}^p(\mathbb{T}^d)$, *its Fourier series coefficients* $|\tilde{f}_m| = \mathcal{O}(1/m^p)$.

**Fact 2.** *For any* $f \in \mathcal{H}^p([-1, 1]^d)$, *its Chebyshev expansion* $|\tilde{f}_m| = \mathcal{O}(1/m^p)$.

**Neural operators.** Neural operators parameterize the mapping $\mathcal{G}_\theta : \phi \mapsto u_\theta$ as,

$$\mathcal{G}_\theta := \mathcal{Q} \circ \sigma(\mathcal{W}^{(L)} + \mathcal{K}^{(L)}) \circ \cdots \circ \sigma(\mathcal{W}^{(1)} + \mathcal{K}^{(1)}) \circ \mathcal{P}, \tag{3}$$

where $\sigma$ is a non-linearity. The operator iteratively updates $v_\theta^{(l)} : \Omega \to \mathbb{R}^{d_l}$, where $d_l$ is the hidden dimension of layer $l$. The input layer $v_\theta^{(0)} = \mathcal{P}(\phi)$ and output layer $u_\theta = \mathcal{Q}(v_\theta^{(L)})$ are parameterized by point-wise NNs. The affine operator $\mathcal{W}^{(l)}$ and the kernel integral operator $\mathcal{K}^{(l)}$ are defined as,

$$(\mathcal{W}^{(l)}(v))(x) = W^{(l)}v(x) + b^{(l)}, \quad (\mathcal{K}^{(l)}(v))(x) = \int_\Omega K^{(l)}(x, y)v(y)d\mu(y), \tag{4}$$

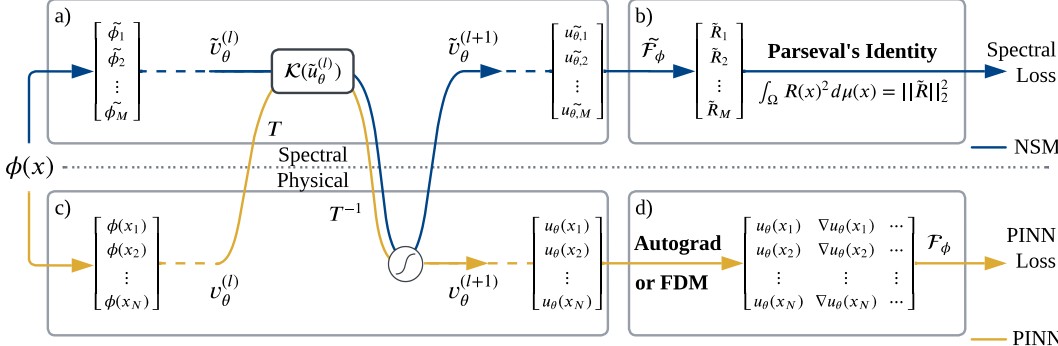

Figure 1: **Schematic of NSM.** We refer to Neural Spectral Methods (NSM) as a general approach to learn PDE solution mappings in the spectral domain. Our method consists of two components: **a)** The parameters $\phi$ are converted to spectral coefficients, $\tilde{\phi}$. In each NN layer $l$, the spectral coefficients $\tilde{v}_\theta^{(l)}$ are transformed by a linear operator $\tilde{\mathcal{K}}$, with the activation $\sigma$ then applied on collocation points in the physical space. **b)** The prediction $\tilde{u}_\theta$ is transformed by $\tilde{\mathcal{F}}_\phi$, the spectral form of the differential operator, which gives the spectral coefficients $\tilde{R}$ of the residual function. The exact residual norm is obtained by Parseval's Identity, giving the spectral loss $||\tilde{R}||_2^2$. We contrast our method against the commonly employed grid-based neural operators with a PINN loss. **c)** General neural operators learn the PDE solutions as transformations of function values on $x_i$. We consider the kernel integral in a more general sense, with transformation $T$ not restricted to a Fourier basis. **d)** Autograd or finite difference methods are used to obtain the higher-order derivatives. The PINN loss is then obtained by approximating the norm of the residual function on the sampled points.

where the input $v$ is in $\Omega \to \mathbb{R}^{d_{l-1}}$ and the outputs are in $\mathbb{R}^{d_l}$. The affine operator and kernel integral operator are used to capture local and non-local transformations, respectively. Given input grid $[x_i]$, general grid-based neural operators parameterize $\mathcal{G}_\theta$ as a mapping between function values:

$$[\phi(x_1) \quad \phi(x_2) \quad \dots \quad \phi(x_N)] \mapsto [u_\theta(x_1) \quad u_\theta(x_2) \quad \dots \quad u_\theta(x_N)]. \qquad (5)$$

## 3.2 Spectral-based neural operators

In this work, we employ the neural operator architecture to model transformations between spectral coefficients. By fixing $\{f_m(x) : x \in \Omega\}_{m \in \mathcal{I}}$ as the chosen basis functions, the solution operator is parameterized as the mapping between the coefficients of the parameter functions $\phi$ and the predicted solutions $u_\theta$. Suppose the series is truncated to $M$ terms, then $\mathcal{G}_\theta$ is parameterized in the spectral domain as:

$$\phi(x) = \sum_{m \in [M]} \tilde{\phi}_m f_m(x) \mapsto u_\theta(x) = \sum_{m \in [M]} \tilde{u}_{\theta,m} f_m(x), \qquad (6)$$

where $\tilde{\phi}$ and $\tilde{u}_\theta$ are the spectral expansion of $\phi$ and $u_\theta$ under the basis $f$. For each layer $l$, the function $v_\theta^{(l)}$ and kernel $K^{(l)}$ are also parameterized under the same basis functions,

$$v_\theta^{(l)}(x) = \sum_{m \in [M]} \tilde{v}_{\theta,m}^{(l)} f_m(x), \quad K^{(l)}(x,y) = \sum_{m \in [M]^2} \tilde{K}_m^{(l)} f_{m_1}(x) f_{m_2}(y), \qquad (7)$$

where $\tilde{v}_\theta^{(l)} \in \mathbb{R}^{M \times d_l}$ and $\tilde{K}^{(l)} \in \mathbb{R}^{M \times M \times d_l \times d_{l-1}}$ are coefficients in the spectral domain. Due to orthogonality, integral transformations $\mathcal{K}^{(l)}$ are actually equivalent to tensor contractions $\tilde{v}_{\theta,m_2 i}^{(l-1)} \cdot \tilde{K}_{m_1 m_2 ij}^{(l)}$. Similarly, affine transformations $\mathcal{W}^{(l)}$ are equivalent to $\tilde{v}_{\theta,mi}^{(l-1)} \cdot W_{ij}^{(l)} + b_{mj}^{(l)}$.

**Non-linear activation functions.** The activation $\sigma$ is applied on the collocation points. We denote $\mathcal{T}$ as the interpolation of function values at collocation points aligned with the basis, and $\mathcal{T}^{-1}$ as the function value evaluation on those collocation points. Then $\tilde{\sigma}$, the spectral counterpart of the activation function $\sigma$, is given by:

$$\tilde{\sigma} = \mathcal{T} \circ \sigma \circ \mathcal{T}^{-1}. \qquad (8)$$

**Aliasing error.**   General grid-based neural operators are prone to aliasing error. When trained and tested on different grid resolutions, the interpolation error is inconsistent, leading to an increased error on the test grids that are a different resolution from the training grid. In contrast, our spectral-based approach circumvents aliasing error. By operating exclusively on fixed collocation points, the interpolation error remains consistent across different input resolutions. This also ensures that the model's computation cost and predictions are resolution-independent.

**Kernel approximation.**   Within each layer, the computational cost of the kernel integral is quadratic. To mitigate this cost, the kernel can be confined to a low-rank or simplified form. Here, we employ FNO's approach, which simplifies the kernel integral into a convolution through a Fourier transformation. More broadly, the kernel is restricted to a diagonal form, $\tilde{K}^{(l)} \in \mathbb{R}^{M \times d_l \times d_{l-1}}$, and,

$$K^{(l)}(x, y) = \sum_{m \in [M]} \tilde{K}_m^{(l)} f_m(x) f_m(y). \tag{9}$$

## 3.3   SPECTRAL TRAINING

After our base NN architecture predicts the solution $\tilde{u}$ in Eq. 6, we train our model using a spectral training procedure. Here, we describe the details of our spectral training procedure.

**Spectral form of the residual function.**   We can derive the exact residual function using the spectral representation of the prediction, denoted by $\tilde{u}$. The spectral representation has a direct correspondence between operations performed on function values and on spectral coefficients. Additionally, differentiation and integration are transformed to algebraic operations on the coefficients. Given the PDE operator $\mathcal{F}_\phi$, we convert it to its spectral correspondence $\tilde{\mathcal{F}}_\phi : \tilde{u}_\theta \mapsto \tilde{R}$, such that,

$$\mathcal{F}_\phi(u_\theta(x)) = \sum_{m \in \mathcal{I}} \tilde{R}_m f_m(x), \tag{10}$$

where $\tilde{R}$ represents the spectral form of the PDE residual function. We describe in more detail how to obtain $\tilde{\mathcal{F}}_\phi$ from $\mathcal{F}_\phi$ for typical nonlinear operators and bases composed of Fourier series and Chebyshev polynomials in §A.1.

**Spectral loss.**   After computing the residual function, we aim to minimize our *spectral loss*, $||\tilde{R}||_2^2$. This method involves projecting the residual function onto a subspace spanned by truncated basis functions, known as the Galerkin projection. The orthogonality of the basis functions is crucial to this procedure. Leveraging Parseval's Identity, we can equate the spectral loss to the weighted norm of the residual function, as outlined below:

**Theorem 1** (Parseval's Identity). *For $R(x) = \sum_{m \in \mathcal{I}} \tilde{R}_m f_m(x)$, we have,*

$$\int_\Omega R(x)^2 d\mu(x) = \sum_{m \in \mathcal{I}} \tilde{R}_m^2. \tag{11}$$

**Contrast with the PINN loss.**   As previously discussed, the PINN loss is obtained by sampling a substantial number of points within the interior domain, followed by employing a numerical quadrature to approximate the integral of $R(x)^2$. This requires differentiating through the NN, which can be computationally expensive, especially for higher-order derivatives, and when a large number of points are sampled to ensure accurate quadrature. For the grid-based PINN loss, which commonly uses finite difference methods to approximate the derivatives, we have provided an error analysis in §B.1. From a theoretical perspective, we show that as long as the grid spacing is finite, the expected solution error can be non-zero, even if the grid-based PINN loss is minimized arbitrarily well.

We bypass this process by using the spectral representation of the solution, and apply spectral transformations to represent the residual function in the same basis. This greatly simplifies the entire optimization procedure, and as we will demonstrate, significantly reduces training time. Note that even though the spectral loss is a weighted norm, the corresponding PINN loss can also be readily constrained:

**Corollary 1.** *For $\frac{dx}{d\mu} \in \mathcal{L}^\infty(\Omega)$, $||R(x)||_{L^2(\Omega)} = \mathcal{O}(||\tilde{R}||_2^2)$.*

This result follows directly by Hölder's inequality. Both Fourier series and Chebyshev polynomials fulfill this condition, ensuring that minimizing the spectral loss also minimizes the PINN loss.

## 4 EXPERIMENTAL RESULTS

We compare NSM to different neural operators with diffeerent loss functions (PINN and spectral losses) on several PDEs: 2D Poission (§4.1), 1D Reaction-Diffusion (§4.2), and 2D Navier-Stokes (§4.3) with both forced and unforced flow. NSM is consistently the most accurate method, and orders of magnitudes faster during both training and inference, especially on large grid sizes.

**Problem setting.** For all experiments, we focus on the *data-constrained setting*, using no interior domain solution data during training (i.e., we train only by minimizing the PDE residual). The task is to learn the mapping from PDE parameters $\phi$ to solutions $u$. During training, each model is given $\mathcal{F}_\phi$ and the parameters $\phi_i$, which are independently sampled in every training iteration.

Recall that the PINN loss (Eq. 12) and the spectral loss (Eq. 13) are used on grid-based and spectral-based models, respectively. For the PINN loss, higher-order derivatives are computed using the finite difference method on a fixed rectangular grid $[x_1 \; x_2 \; \ldots \; x_N]$. For our spectral loss, the $M$-term residual series is directly transformed from the predicted solution $\tilde{u}_\theta$.



PINN Loss

Spectral Loss



$$\frac{1}{|\{\phi_i\}|} \sum_{\phi_i} \frac{1}{N} \sum_{n \in [N]} F_{\phi_i}(u_{\theta,i}(x_n), \nabla u_{\theta,i}(x_n)...)^2 \quad (12) \qquad \frac{1}{|\{\phi_i\}|} \sum_{\phi_i} \sum_{m \in [M]} \tilde{\mathcal{F}}_{\phi_i}(\tilde{u}_{\theta,i})_m^2 \quad (13)$$

For each problem, the test set consists of $N = 128$ PDE parameters, denoted by $\phi_i$. Each $\phi_i$ is sampled from the same distribution used at training time, and $u_i$ is the corresponding reference solution. For each prediction $u_\theta$, we evaluate two metrics: $L_2$ relative error $||u_{\theta,i} - u_i||_2/||u_i||_2$ and the PDE residual error $||\mathcal{F}_\phi(u_{\theta,i})||_2$. Both metrics are computed on the test set resolution, and averaged over the dataset. Additional details about the experimental setup are provided in §C.

We include the following models for comparison:

- **FNO + PINN loss.** A grid-based FNO architecture (Li et al., 2020), trained with a PINN loss. The model is trained on different grid sizes, which are indicated by the corresponding labels (e.g., FNO$\times 64^2$ means a grid size of $64 \times 64$ is used to calculate the PINN loss).
- **SNO + Spectral loss.** Ablation model: A base SNO architecture (Fanaskov & Oseledets, 2022), trained with our spectral loss.
- **T1 + PINN loss / Spectral loss.** Ablation model: A base TransformOnce architecture (Poli et al., 2022), trained with either a PINN loss or our spectral loss.
- **CNO + PINN loss (ours).** Ablation model: The base architecture is identical to NSM, but trained with a PINN loss on the largest grid size used by FNO, i.e. 256 on each dimension.
- **NSM (ours).** Our proposed spectral-based neural operator using Fourier and Chebyshev basis on periodic and non-periodic dimensions, and it is trained with our spectral loss.

For a fair comparison, the base architecture, number of parameters, and other hyperparameters are kept exactly the same across neural operator-based models. Detailed parameters are provided in §C.

### 4.1 POISSON EQUATION

We study the 2D Poisson equation, $-\Delta u(x, y) = s(x, y)$, $x, y \in \mathbb{T}$, with periodic boundary conditions. The source term $s$ is sampled from a random field with a length scale of $0.2$. The task is to learn the mapping from $s$ to the potential field $u$. We evaluate the predictions using an $L_2$ relative error with respect to the analytical solution. Additional details and results for Dirichlet boundary conditions are in §C.1.

Table 1: $L_2$ relative error (%) for the periodic Poission equation.

| SNO+Spectral | T1$\times 64^2$+PINN | T1+Spectral | FNO$\times 64^2$+PINN | FNO$\times 128^2$+PINN | FNO$\times 256^2$+PINN | NSM (ours) |
|---|---|---|---|---|---|---|
| $0.59 \pm 0.12$ | $3.22 \pm 0.49$ | $0.302 \pm 0.071$ | $4.24 \pm 0.13$ | $2.01 \pm 0.03$ | $1.75 \pm 0.02$ | $\mathbf{.057 \pm .012}$ |

**Results.** The results are summarized in Tab. 1. Since the solution operator is an inverse Laplacian, all models can theoretically express it with one layer (see discussions in C.1), but the FNO + PINN models exhibits high error, even when trained with large grids and for a longer training time. This simple example highlights the inherent optimization challenges with the grid-based PINN loss.

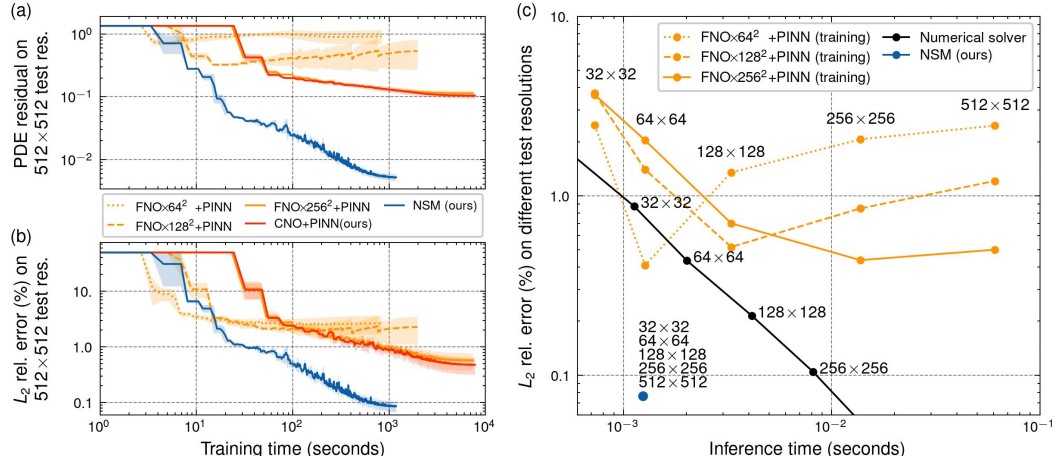

Figure 2: **Reaction-Diffusion equation with $\nu = 0.01$.** In **a)** and **b)**, the $L_2$ relative error and PDE residual on the test set are plotted over training each model. The grid-based methods (FNO trained with a PINN loss) show improved accuracy as the grid resolution increases, but are significantly slower to train. When tested on different resolutions, significant aliasing errors occur on test grid resolutions that differ from the training grid resolution. In contrast, NSM has a much low error and PDE residual, and achieves this lower error $100\times$ faster than the grid-based methods. In **c)**, when compared with iterative numerical solvers on different resolutions, NSM achieves the same level of accuracy with a $10\times$ speedup. Notably, both the accuracy and computational cost of NSM remains constant, regardless of grid resolution.

## 4.2 REACTION-DIFFUSION EQUATION

We study the 1D periodic Reaction-Diffusion system with different diffusion coefficients:

$$
\begin{aligned}
u_t - \nu u_{xx} &= \rho u(1-u), & x \in \mathbb{T},\ t \in [0, T], \\
u(x, 0) &= h(x), & x \in \mathbb{T},
\end{aligned}
\tag{14}
$$

where $\nu$ is the diffusion coefficient, and $h(x)$ is the initial condition sampled from a random field with a length scale of $0.2$. Given $h(x)$, the model learns to predict $u(x, t)$ up to time $T = 1$. The initial condition is enforced by transforming the prediction $u(x, t)$ to $u(x, t) \cdot t + h(x)$.

Table 2: $L_2$ relative error (%) and computation cost (GFLOP) for the Reaction-Diffusion equation.

|  | SNO+Spectral | FNO×$64^2$+PINN | FNO×$128^2$+PINN | FNO×$256^2$+PINN | CNO+PINN (ours) | NSM (ours) |
|---|---|---|---|---|---|---|
| $\nu = 0.005$ | $4.56 \pm 0.99$ | $2.30 \pm 0.19$ | $0.94 \pm 0.11$ | $0.33 \pm 0.04$ | $0.20 \pm 0.01$ | $\mathbf{.075 \pm .016}$ |
| $\nu = 0.01$ | $5.41 \pm 4.43$ | $2.64 \pm 0.97$ | $2.27 \pm 1.19$ | $0.57 \pm 0.19$ | $0.48 \pm 0.16$ | $\mathbf{.086 \pm .019}$ |
| $\nu = 0.05$ | $87.76 \pm 52$ | $11.82 \pm 5.4$ | $3.25 \pm 1.29$ | $1.06 \pm 0.28$ | $0.78 \pm 0.01$ | $\mathbf{.083 \pm .006}$ |
| $\nu = 0.1$ | $152.8 \pm 58$ | $13.03 \pm 6.4$ | $4.90 \pm 2.40$ | $4.07 \pm 2.00$ | $1.28 \pm 0.42$ | $\mathbf{.077 \pm .005}$ |
| Train/Test | $0.91/0.03$ | $7.30/15.8$ | $31.3/15.8$ | $138.5/15.8$ | $200.3/25.0$ | $15.0/0.32$ |

**Results.** The main results are summarized in Tab. 2, including the results for TransformOnce (Poli et al., 2022) in §C.2. The training curves for the PDE residual and relative error on the test set for diffusion coefficient $\nu = 0.01$ are shown in Fig. 2. The grid-based models using the PINN loss improve in relative error with a higher-resolution grid, but require significantly longer training time. In contrast, NSM consistently achieves high accuracy while maintaining a low computation cost. As the diffusion coefficient increases, NSM shows strong robustness and consistently achieves low solution error, while the other models all increase significantly in solution error.

We also compare the ML models to a standard numerical solver (Simpson & Landman, 2006), as shown in Fig. 2c. During inference time, the accuracy and computational cost of NSM remains constant, regardless of the spatiotemporal resolution of the domain. NSM exhibits a $10\times$ increase in speed when compared to a numerical solver with a comparable level of accuracy. Solution error distributions and additional details are in §C.2.

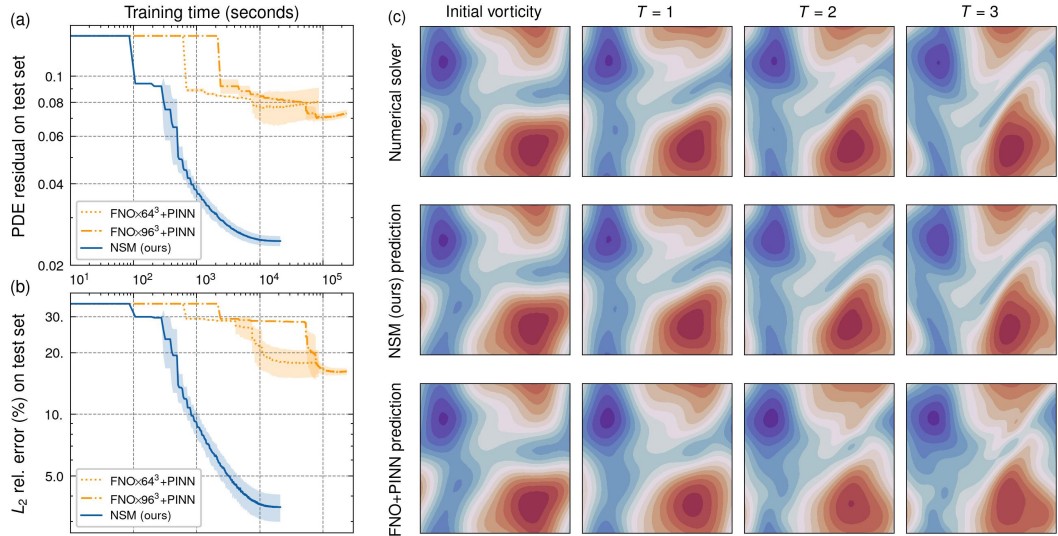

Figure 3: **Navier-Stokes equation with** $\nu = 10^{-4}$. In **a)** and **b)**, the relative error and PDE residual on the test set are plotted. NSM achieves low $L_2$ error and PDE residual $100\times$ faster than FNO + PINN methods, and is an order of magnitude more accurate. **c)** NSM captures fine features of the vorticity evolution accurately, while the grid-based approach fails to predict the overall shape.

### 4.3 NAVIER-STOKES EQUATION

We study the vorticity form of the 2D periodic Navier-Stokes (NS) equations:

$$
\begin{aligned}
\partial_t w + u \cdot \nabla w &= \nu \Delta w + f, & x \in \mathbb{T}^2,\ t \in [0, T], && \text{where} && \nabla \cdot u = 0, \\
w(x, 0) &= w_0(x), & x \in \mathbb{T}^2, &&&& \nabla \times u = w,
\end{aligned}
\tag{15}
$$

where $\nu$ is the viscosity, $w$ is the vorticity field, and $f$ is the forcing term. The initial vorticity $w_0$ is sampled from a random field with a length scale of $0.8$. Given $w$, the velocity field $u$ is determined by applying the inverse Laplacian operator. The model is trained to learn the evolution of vorticity.

We first consider the unforced flow with different $\nu$ values and $T = 3s$. The solution is diffusive for large $\nu$, and becomes more challenging to learn as the viscosity decreases, due to the sharp features in the solution. For FNO trained with the PINN loss, using a grid resolution larger than 96 becomes intractable to train, due to the cost of compute and memory. The results are summarized in Tab. 3 and the case for $\nu = 10^{-4}$ is shown in Fig. 3. NSM significantly outperforms grid-based FNO with the PINN loss in terms of both error and computational speed, achieving accurate results $100\times$ faster (see Fig. 3a and Fig. 3b).

Table 3: $L_2$ relative error (%) for unforced NS equations.

|  | FNO$\times 64^3$+PINN | FNO$\times 96^3$+PINN | NSM (ours) |
|---|---|---|---|
| $\nu = 10^{-2}$ | $8.18 \pm 2.83$ | $7.90 \pm 0.57$ | $\mathbf{0.71 \pm 0.02}$ |
| $\nu = 10^{-3}$ | $14.81 \pm 0.67$ | $11.99 \pm 0.86$ | $\mathbf{1.65 \pm 0.26}$ |
| $\nu = 10^{-4}$ | $17.88 \pm 2.67$ | $16.20 \pm 0.61$ | $\mathbf{3.53 \pm 0.53}$ |
| Train | 41h | 131h | 10h |

Next, we consider the long temporal transient flow under the forcing term $f = 0.1(\sin(2\pi(x + y)) + \cos(2\pi(x + y)))$ and $T = 50s$, following the setting in Li et al. (2020; 2021). This is a significantly more challenging task, as it requires propagating the initial condition over an extended time interval. Nevertheless, as summarized in Tab. 4, NSM maintains high accuracy, while grid-based FNO with the PINN loss collapses during training and fails entirely. Further details for both unforced and forced flow can be found in §C.3.

Table 4: $L_2$ relative error (%) for the long temporal transient NS equation.

|  | FNO$\times 96^3$+PINN | NSM (ours) |
|---|---|---|
| $\nu = 1/500$ | $55.1 \pm 17.4$ | $\mathbf{13.2 \pm 0.57}$ |

**Conclusion.** We introduce an ML approach for solving PDEs, inspired by spectral methods. By utilizing orthogonal basis functions and their spectral properties, we demonstrate numerous advantages for learning PDEs in the spectral domain. Our method is evaluated on different PDEs, and achieves significantly lower error and increased efficiency when compared to current ML methods.

**Acknowledgements.** This work was supported by the U.S. Department of Energy, Office of Science, Office of Advanced Scientific Computing Research, Scientific Discovery through Advanced Computing (SciDAC) program under contract No. DE-AC02-05CH11231. It was also supported in part by the Office of Naval Research (ONR) under grant N00014-23-1-2587. We also acknowledge generous support from Google Cloud and AWS Cloud Credit for Research. We thank Rasmus Malik Høegh Lindrup and Sanjeev Raja for helpful discussions and feedback.

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

## A   ORTHOGONAL BASIS

**Definition 1** (Orthogonal basis). *A set of real-valued functions $\{f_m(x) : x \in \Omega\}_{m \in \mathcal{I}}$ is said to be orthogonal w.r.t a probability measure $\mu : \Omega \to \mathbb{R}^+$, if for any two indices $i, j \in \mathcal{I}$ we have,*

$$\int_\Omega f_i(x) f_j(x) d\mu(x) = \mu_i \delta_{ij}, \tag{16}$$

*where constants $\mu_i \in \mathbb{R}^+$ and $\delta_{ij} = 1_{i=j}$. If $\mu_i \equiv 1$, they are also orthonormal. Furthermore, when their finite linear combinations are dense in $\mathcal{L}^2(\Omega; \mu)$, they are called an orthogonal basis.*

Given that a set of orthogonal basis functions can be easily normalized, we assume their orthonormality unless otherwise specified in this paper. For multi-dimensional problems, we can use the cross product of basis functions on each dimension to form another orthogonal basis:

**Proposition 1.** *Given orthogonal basis $\{f_m^{(1)}(x_1) : x_1 \in \Omega_1\}_{m \in \mathcal{I}_1}$ and $\{f_m^{(2)}(x_2) : x_2 \in \Omega_2\}_{m \in \mathcal{I}_2}$ in $\mathcal{L}^2(\Omega_1; \mu_1)$ and $\mathcal{L}^2(\Omega_2; \mu_2)$, the cross product $\{f_{m_1}^{(1)}(x_1) f_{m_2}^{(2)}(x_2) : x \in \Omega_1 \times \Omega_2\}_{m \in \mathcal{I}_1 \times \mathcal{I}_2}$ is an orthongonal basis in the product space $\mathcal{L}^2(\Omega_1 \times \Omega_2; \mu_1 \times \mu_2)$.*

### A.1   SPECTRAL CORRESPONDENCE

This section details operations in common PDE operators and their spectral correspondence with Fourier and Chebyshev[1] bases. While operations on Fourier bases are well-known, for an in-depth introduction of Chebyshev polynomials, see Mason & Handscomb (2002).

Let $u, v$ be the 1-dimensional function in consideration. Denote $\tilde{u}, \tilde{v} \in \mathbb{R}^M$ as their $M$-term expansion under a Fourier or Chebyshev basis. We have the following properties in Tab. 5:

Table 5: Spectral correspondence of common operations.

| | Fourier basis | Chebyshev polynomials |
|---|---|---|
| $f$ | $f_{m,\cos} = \cos(2\pi m x)$ $f_{m,\sin} = \sin(2\pi m x)$ | $f_m(x) = \cos(m \cos^{-1}(x))$ |
| $f = u+v$ | $\tilde{f} = \tilde{u} + \tilde{v}$ | $\tilde{f} = \tilde{u} + \tilde{v}$ |
| $f = u \cdot v$ | $\tilde{f}_{m,\cos} = \sum\limits_{n \le m, M} \tilde{u}_{n,\cos}\tilde{v}_{m-n,\cos} - \tilde{u}_{n,\sin}\tilde{v}_{m-n,\sin}$ $\tilde{f}_{m,\sin} = \sum\limits_{n \le m, M} \tilde{u}_{n,\cos}\tilde{v}_{m-n,\sin} + \tilde{u}_{n,\sin}\tilde{v}_{m-n,\cos}$ | $\tilde{f}_m = \sum\limits_{n \le m, M} \tilde{u}_n \tilde{v}_{m-n} + \sum\limits_{n \le M-m} \tilde{u}_m \tilde{v}_{m+n} + \tilde{u}_{m+n}\tilde{v}_m$ |
| $f = \frac{du}{dx}$ | $\tilde{f}_{m,\cos} = 2\pi m \tilde{u}_{m,\sin}$ $\tilde{f}_{m,\sin} = -2\pi m \tilde{u}_{m,\cos}$ | $\tilde{f}_m = \sum\limits_{2n \le M-m} 2(m+2n+1)\tilde{u}_{m+2n+1}$ |

Note that sums in convolution form $\sum_l a_l b_{k-l}$ can be accelerated by Fast Fourier Transforms (FFT). For multi-dimensional problems, the above operations can be done on each dimension sequentially.

## B   GRID-BASED METHODS

**Complexity of computing residuals.**   We continue the discussion in our introduction and Li et al. (2021), where we look at the complexity of computing exact derivatives for grid-based neural operators. Consider differentiating a parameterized kernel $\mathcal{K}$:

$$v(x) = (\mathcal{K}u)(x) = \int_\Omega K(x, y) u(y) d\mu(y) = \sum_{m \in \mathcal{I}^2} \int_\Omega \tilde{K}_m f_{m_1}(x) f_{m_2}(y) u(y) d\mu(y). \tag{17}$$

For a given set of sampled points $\{x_i\}_{i=1}^N$, suppose both $u$ and $v$ are queried at these points. Then, regardless of how the integral is implemented or approximated, generally we have $\frac{\partial v(x_i)}{\partial u(x_j)} \ne 0$ for every $i, j \in [N]$. Therefore, the complexity to obtain the exact derivatives is quadratic in $N$.

---

[1]For implementation purposes, we use shifted Chebyshev polynomial on $[0, 1]$ throughout this work, i.e. we consider functions $T_m^*(x) = T_m(2x - 1)$, where $T_m$ is the standard Chebyshev polynomial on $[-1, 1]$.

## B.1 Error analysis for grid-based PINN loss

Grid-based methods estimate residuals on a discretized domain. As grid spacings (e.g., $\Delta x$, $\Delta t$) approach zero, the estimated residuals tend towards being exact. However, the grid size is limited by the expensive backpropagation in the training time, causing discretization errors. In this section, we demonstrate this with an error analysis of the grid-based PINN loss for elliptic PDEs (Evans, 2022).

**Problem setting.** Consider an operator $\mathcal{L}$ on a bounded open subset $\Omega \subset \mathbb{R}^d$:

$$\mathcal{L}u = -\nabla \cdot (A\nabla u), \tag{18}$$

where $A(x) \in \mathbb{R}^{d \times d}$ are coefficient functions. In order to guarantee the uniqueness of the solution, we assume $A$ satisfies the following uniformly elliptic condition for almost every $x$ in $\Omega$:

$$s||\xi||_2^2 \leq \xi^T A(x)\xi \leq S||\xi||_2^2 \quad \text{for all } \xi \in \mathbb{R}^d, \tag{19}$$

where $s, S > 0$ are absolute constants. Given $\phi \in \mathcal{L}^2(\Omega)$, consider the Dirichlet problem $\mathcal{L}u_\phi = \phi$ for $u_\phi \in \mathcal{H}^2(\Omega)$ satisfying $u_\phi = 0$ on $\partial\Omega$. The task is to learn the transformation $\mathcal{G} : \phi \mapsto u_\phi$.

**Error analysis.** Given an input $\phi$, denote the model prediction as $\bar{u}_\phi$. Consider minimizing the grid-based PINN loss using finite difference methods with grid spacing $h > 0$, i.e., the discrete gradient operator $\nabla_h$ is used to approximate $\nabla$ in $\mathcal{L}$. Suppose that the grid-based PINN loss is minimized, that is for some small $\epsilon, \delta > 0$, the probability is,

$$\mathbb{P}(||\nabla_h \cdot (A\nabla_h \bar{u}_\phi) + \phi||_{\mathcal{L}^2(\Omega)} \leq \epsilon) \geq 1 - \delta. \tag{20}$$

**Proposition 2.** *Under the assumptions of Eq. 19 and Eq. 20, the estimation for the expected solution error holds:*

$$\mathbb{E}[||\bar{u}_\phi - u_\phi||_{\mathcal{L}^2(\Omega)}] \geq \frac{s||\nabla_h(u_\phi - \bar{u}_\phi)||_{\mathcal{L}^2(\Omega)}^2 + s||\nabla u_\phi||_{\mathcal{L}^2(\Omega)}^2 - S||\nabla_h\bar{u}_\phi||_{\mathcal{L}^2(\Omega)}^2 - \epsilon}{||\nabla_h \cdot (A\nabla_h u_\phi)||_{\mathcal{L}^2(\Omega)}/(1-\delta)}. \tag{21}$$

*Proof.* Consider the inner product $\langle \nabla_h \cdot (A\nabla_h u_\phi), \bar{u}_\phi - u_\phi \rangle_\Omega$. By Cauchy-Schwartz:

$$\langle \nabla_h \cdot A(\nabla_h u_\phi), u_\phi - \bar{u}_\phi \rangle_\Omega \leq ||\nabla_h \cdot (A\nabla_h u_\phi)||_{\mathcal{L}^2(\Omega)} \cdot ||\bar{u}_\phi - u_\phi||_{\mathcal{L}^2(\Omega)}. \tag{22}$$

We can also use integration by parts, and then this inner product equals:

$$
\begin{aligned}
&\int_\Omega (\bar{u}_\phi - u_\phi)\nabla_h \cdot (A\nabla_h u_\phi)dx, \\
=\ &\int_\Omega \nabla_h(u_\phi - \bar{u}_\phi) \cdot (A\nabla_h(u_\phi - \bar{u}_\phi)) + (\bar{u}_\phi - u_\phi)\nabla_h \cdot (A\nabla_h\bar{u}_\phi)dx, \\
=\ &\int_\Omega \nabla_h(u_\phi - \bar{u}_\phi) \cdot (A\nabla_h(u_\phi - \bar{u}_\phi))dx - \int_\Omega \nabla_h\bar{u}_\phi \cdot (A\nabla_h\bar{u}_\phi)dx - \int_\Omega u_\phi\nabla_h \cdot (A\nabla_h\bar{u}_\phi)dx.
\end{aligned} \tag{23}
$$

By the uniformity condition in Eq. 19, the first two terms can be estimated by:

$$
\begin{aligned}
\int_\Omega \nabla_h(u_\phi - \bar{u}_\phi) \cdot (A\nabla_h(u_\phi - \bar{u}_\phi))dx &\geq s||\nabla_h(u_\phi - \bar{u}_\phi)||_{\mathcal{L}^2(\Omega)}^2, \\
\int_\Omega \nabla_h\bar{u}_\phi \cdot (A\nabla_h\bar{u}_\phi)dx &\leq S||\nabla_h\bar{u}_\phi||_{\mathcal{L}^2(\Omega)}^2.
\end{aligned} \tag{24}
$$

By assumption Eq. 20, the following holds with probability $1 - \delta$:

$$\int_\Omega u_\phi\nabla_h \cdot (A\nabla_h\bar{u}_\phi)dx \leq \int_\Omega u_\phi\nabla \cdot (A\nabla u_\phi)dx + \epsilon \leq -s||\nabla u_\phi||_{\mathcal{L}^2(\Omega)}^2 + \epsilon. \tag{25}$$

Then, with high probability, the inner product in Eq. 22 can be estimated by $\mathbb{P}(\langle \nabla_h \cdot (A\nabla_h u_\phi), \bar{u}_\phi - u_\phi \rangle_\Omega \geq s||\nabla_h(u_\phi - \bar{u}_\phi)||_{\mathcal{L}^2(\Omega)}^2 + s||\nabla u_\phi||_{\mathcal{L}^2(\Omega)}^2 - S||\nabla_h\bar{u}_\phi||_{\mathcal{L}^2(\Omega)}^2 - \epsilon) \geq 1 - \delta$. Rearranging the terms, the desired result follows by Markov's inequality. $\square$

**Discussion.** This analysis implies that when the difference between the exact derivatives $\nabla u_\phi$, and the approximated derivatives $\nabla_h u_\phi$, is sufficiently large, there exists some prediction $\bar{u}_\phi$, such that the solution error for the prediction that minimizes the grid-based PINN loss can also be large. The spectral loss has no such issue, as it is able to always obtain exact residuals.

## C  EXPERIMENT SETUP AND DETAILS

Our code is publicly available at `https://github.com/ASK-Berkeley/Neural-Spectral-Methods`. All experiments are implemented using the JAX framework (Bradbury et al., 2018).

**Random fields.** We use random fields for the distribution of PDE parameters, such as initial conditions or coefficient fields. For random fields associated with periodic boundaries, we employ a periodic kernel $k(x, x') = \exp\left(-\frac{\sin^2\left(\frac{\pi}{2}|x-x'|\right)}{l^2/2}\right)$ to sample values on a uniform grid and use a Fourier basis for interpolation. Otherwise, we employ a RBF kernel $k(x, x') = \exp\left(-\frac{|x-x'|^2}{2l^2}\right)$ to sample values on Chebyshev collocation points and use Chebyshev polynomials for interpolation.

**Training.** During training, the PDE parameters are sampled online from random fields, as describe above. We use a batch size of 16. The learning rate is initialized to $10^{-3}$, and exponentially decays to $10^{-6}$ through the whole training. Experiments are run with 4 different random seeds, and averaged over the seeds.

**Testing.** For each problem, the test set consists of $N = 128$ parameters sampled from the same distribution used for training (different from the training samples). The reference solutions are generated as specified in §C.1, §C.2, and §C.3. Both $L_2$ relative error and PDE residuals are averaged over the whole test set.

**Profiling.** During training or inference, the wall clock time is measured by the average elapsed time during 16 calls of just-in-time compiled gradient descent step or a forward pass. The computation cost of numerical solvers in the Reaction-Diffusion problem is measured similarly. The parameter countings for each model used in each experiment are provided in Tab. 6.

Table 6: Model size (number of trainable parameters) in each experiment.

| PDE | Poisson equation | | Reaction-Diffusion equation | | | | Navier-Stokes equation | | Forced NS |
|---|---|---|---|---|---|---|---|---|---|
| Model | SNO | FNO / NSM | SNO | CNO | FNO / NSM | T1 | FNO / NSM | T1 | FNO / NSM |
| # Param. | 7.5M | 31.6M | 33.6M | 67.2M | 67.2M | 167.8M | 236.2M | 782.4M | 82.7M |

**Boundary conditions.** Similar to classic spectral methods, the imposition of boundary and initial conditions is straightforward. Periodic boundaries can be automatically satisfied by using a Fourier basis, and other types of boundary conditions can be enforced by applying a mollifier to the coefficients. This procedure can be specific to the problem at hand, and we show examples in §4.

### C.1  POISSON EQUATION

Besides the periodic Poisson equation, we also show results for Dirichlet boundary conditions.

**Parameters.** For periodic boundary conditions, all models use Fourier bases in $x$ and $y$ dimensions. For Dirichlet boundary conditions, both SNO and NSM use Chebyshev polynomials in both dimensions. For each model, we use 4 layers with 64 hidden dimensions. In each layer, we use 31 modes in both dimensions. All models are trained for 30k steps, except for FNO with a grid size of 256, which requires 100k steps to converge. For the PINN loss, the discrete Laplacian is computed with a 5-point stencil, and with a periodic wrapping for the periodic boundary case. The analytic solutions are evaluated on resolution $256 \times 256$.

All models use ReLU activations, except those using a PINN loss which totally collapse during the training. Therefore, we use $\tanh$ activations for FNO+PINN and T1+PINN, and report these results.

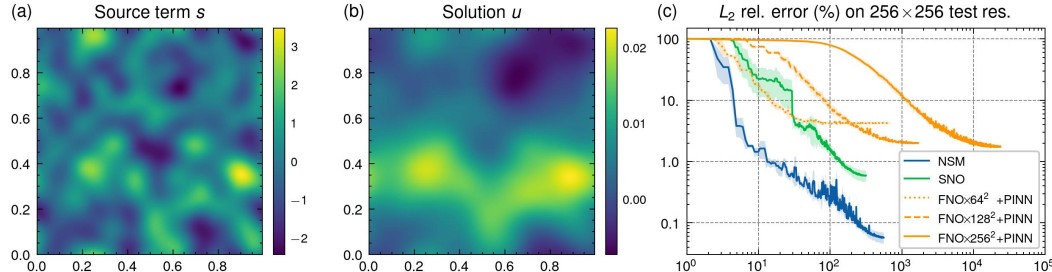

Figure 4: **(a)** Source term **(b)** Solution of periodic Poisson equation. An additional constraint $u(0,0) = 0$ is imposed to ensure the uniqueness of the solution. **(c)** Relative error curves during training. NSM converges significantly faster and to a lower error than the other methods.

Table 7: $L_2$ relative error (%) of the 2D Poission equation with Dirichlet boundary conditions.

| SNO+Spectral | T1×64²+PINN | T1+Spectral | FNO×64²+PINN | FNO×128²+PINN | FNO×256²+PINN | NSM (ours) |
|---|---|---|---|---|---|---|
| $39.9 \pm 36.6$ | $2.24 \pm 1.61$ | $0.407 \pm 0.068$ | $3.52 \pm 0.29$ | $3.61 \pm 0.27$ | $0.443 \pm 0.032$ | $\mathbf{0.280 \pm 0.017}$ |

**Results.** The $L_2$ relative error for both types of boundaries are summarized in Tab. 1 and Tab. 7, and the case for periodic boundary conditions is shown in Fig. 4. For the periodic case, the inverse Laplacian operator is a convolution with $g(x) \propto ||x||_2$. Therefore, all models can exactly represent the solution with one layer. However, FNOs with a PINN loss fail to achieve an reasonable level of accuracy. As the grid size increases, error improves but it requires more training steps to converge.

For Dirichlet boundary conditions, the solution operator is no longer diagonal under either Fourier or Chebyshev basis. Nevertheless, NSM still achieves the highest accuracy among each models.

### C.2 REACTION-DIFFUSION EQUATION

**Parameters.** Both CNO and NSM use Chebyshev polynomials in the $t$ dimension, and Fourier bases in the $x$ dimension. All models use ReLU activations. For each model, we use 4 layers with 64 hidden dimensions. In each layer, we use 32 modes in the time dimension and 64 modes in the spatial dimension. For $\nu = 0.005$ and $\nu = 0.01$, all models are trained for 30k steps. Due to the increasing problem difficulty as $\nu$ becomes larger, the training steps are increased to 100k for $\nu = 0.05$, and 200k for $\nu = 0.1$, respectively.

The reference solution is generated by a standard operator splitting method (Simpson & Landman, 2006) with resolution $4096 \times 4096$, and then downsampled to resolution $512 \times 512$.

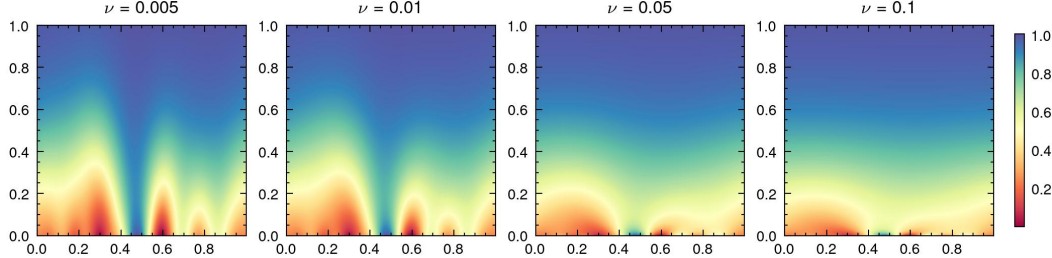

Figure 5: Solutions of the 1D Reaction-Diffusion equation with different[2] diffusion coefficients $\nu$. Following Krishnapriyan et al. (2021), we use a reaction coefficient $\rho = 5$ and four different values for the diffusion coefficient. As $\nu$ increases, the problem becomes progressively harder to solve.

---

[2]The domain on the $x$-axis is taken as $[0, 1)$, unlike Krishnapriyan et al. (2021), which considered the interval $[0, 2\pi)$. Diffusion coefficients are rescaled accordingly for similar solution behaviors. For diffusion coefficients larger than 0.1, the solution becomes almost fully diffusive and constant over the domain.

**Results.** The $L_2$ relative error for all models are summarized in Tab. 2 and Tab. 8, and distributions of absolution error over the test set are shown in Fig. 6. As $\nu$ increases, the error distribution of the grid-based methods significantly increases, while the error of NSM remains almost identical.

Table 8: $L_2$ relative error (%) and computation cost (GFLOP) for the Reaction-Diffusion equation, compared to T1 + Spectral loss and T1 + PINN loss (continued from Tab. 2).

|  | T1+Spectral | T1$\times 64^2$+PINN | T1$\times 128^2$+PINN | T1$\times 256^2$+PINN | CNO+PINN (ours) | NSM (ours) |
|---|---|---|---|---|---|---|
| $\nu = 0.005$ | $0.68 \pm 0.02$ | $2.30 \pm 0.19$ | $0.94 \pm 0.11$ | $0.32 \pm 0.04$ | $0.20 \pm 0.01$ | $\mathbf{.075 \pm .016}$ |
| $\nu = 0.01$ | $0.84 \pm 0.08$ | $2.64 \pm 0.97$ | $2.27 \pm 1.19$ | $0.57 \pm 0.19$ | $0.48 \pm 0.16$ | $\mathbf{.086 \pm .019}$ |
| $\nu = 0.05$ | $41.36 \pm 69$ | $60.10 \pm 8.5$ | $81.87 \pm 3.1$ | $90.86 \pm 6.8$ | $0.78 \pm 0.01$ | $\mathbf{.083 \pm .006}$ |
| $\nu = 0.1$ | $196.5 \pm 0.6$ | $94.91 \pm 23.7$ | $101.7 \pm 2.5$ | $94.89 \pm 5.4$ | $1.28 \pm 0.42$ | $\mathbf{.077 \pm .005}$ |
| Train/Test | $4.08/0.17$ | $6.64/6.35$ | $16.5/6.35$ | $60.1/6.35$ | $200.3/25.0$ | $15.0/0.32$ |

Both SNO and T1 perform non-linear activations directly on spectral coefficients, a design choice that differs from FNO's approach of activating function values. This distinction leads to poor robustness in SNO and T1 for increasing $\nu$ values. In contrast, NSM shows a notable robustness to the increasing diffusivity, validating our decision to apply activations at the collocation points.

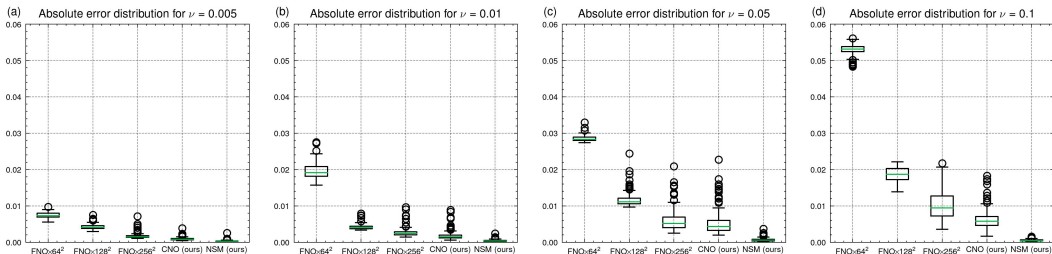

Figure 6: Reaction-Diffusion equation: Error distribution on the test set for each value of $\nu$.

### C.2.1 Ablation study: Fixed-grid training

An insightful observation in our study is the connection between our spectral training and the practice of training grid-based FNOs on fixed-resolution grids, followed by interpolation for inference on different grid sizes. In this section, we provide an ablation study on this fixed-grid training strategy.

**Problem setup.** We use exactly the same PDE problem settings as in §4.2. The ablation model is trained on its collocation points, i.e. a fixed-resolution grid with $32 \times 64$ nodes. During inference time, the model prediction is interpolated using the Fourier basis, and queried at the test resolution.

**Results.** The $L_2$ relative error for this ablation model is summarized in Tab. 9. FNO trained with a fixed-resolution grid fails to converge, showing the important distinction between our spectral training and the practice of fixing training resolution and interpolating.

Table 9: $L_2$ relative error (%) for fixed-grid training.

| $\nu = 0.005$ | $\nu = 0.01$ | $\nu = 0.05$ | $\nu = 0.1$ |
|---|---|---|---|
| $55.7 \pm 0.01$ | $76.9 \pm 0.02$ | $79.5 \pm 0.04$ | $80.4 \pm 0.23$ |

### C.3 Navier-Stokes equation

**Parameters for the unforced flow.** For each model, we use 10 layers with 32 hidden dimensions. In each layer, we use 12 modes in the time dimension and 31 modes in both spatial dimensions.

**Parameters for the forced flow.** For each model, we use 5 layers with 32 hidden dimensions. In each layer, we use 18 modes in the time dimension and 31 modes in both spatial dimensions.

The reference solution is generated numerically using a time-stepping scheme similar to Li et al. (2020), with a spatial resolution of $256 \times 256$. The time stepping interval $\Delta t$ is set to $10^{-3}$ for the unforced flow, and $5 \times 10^{-3}$ for the forced flow. Solution is recorded at a temporal resolution of 64.

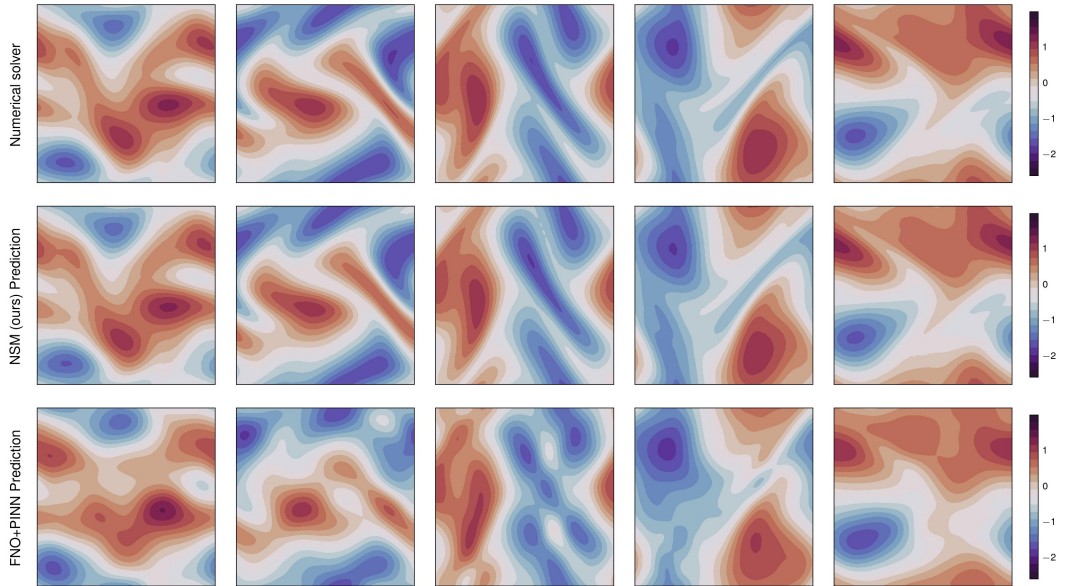

Figure 7: Visualization of unforced flow at time $T = 3$ for viscosity $\nu = 10^{-4}$, using various initial vorticity samples.

NSM uses a Fourier basis in the $x$, $y$ dimensions and Chebyshev polynomials in the $t$ dimension. All models use ReLU activations. All models' learning rate schedulers target 200k steps. For FNO with the PINN loss, the training is limited to 1 day for grid size $64^3$ and 3 days for $96^3$ respectively, after the loss plateaus.

Table 10: $L_2$ relative error (%) for unforced NS equation, compared to T1 + Spectral loss and T1 + PINN loss (continued from Tab. 3).

|  | FNO$\times 64^3$+PINN | T1$\times 64^3$+PINN | T1+Spectral | NSM (ours) |
|---|---|---|---|---|
| $\nu = 10^{-4}$ | $17.88 \pm 2.67$ | $34.6 \pm 0.21$ | $9.22 \pm 0.05$ | $\mathbf{3.53 \pm 0.53}$ |

**Results.** The $L_2$ relative errors for both problems are summarized in Tab. 3, Tab. 4, and Tab. 10, and a visualization of the predicted vorticity for the unforced flow is shown in Fig. 7. FNO with a PINN loss fails to predict both the overall shape and the fine features of the vorticity evolution.

## D    DISCUSSION: LIMITATIONS

NSM applies to a wide range of problems, but is not a one-size-fits-all solution. Similar to those in numerical analysis, efficient spectral representations necessitate sufficient regularities of the solution. Therefore, predicting solutions that involve singularities requires more basis functions. Additionally, NSM is designed for low-dimensional problems, e.g., problems described in $x, y, z$ (and $t$) coordinates. Extensions of our spectral training strategy to high-dimensional problems are left for future research.

