# OpenReview forum: "Neural Spectral Methods: Self-supervised learning in the spectral domain"
_ICLR.cc/2024/Conference — ICLR 2024 poster_

### Official Review · Reviewer_4uLV · 2023-10-25

**Soundness:** 2 fair
**Presentation:** 3 good
**Contribution:** 3 good
**Rating:** 8
**Confidence:** 4

**Summary:**

The authors propose an improvement upon Spectral Neural Operators, essentially proposing to train these architectures in spectral space using Parseval's identity and input data in form of spectral coefficients rather than signals evaluated on grid points. As such, the approach has some novelty and is relevant to the ML+PDE community. The authors show that this approach can outperform existing methods on 3 simple PDE problems (Poisson, Reaction-Diffusion, 2D NS).

**Strengths:**

The paper introduces some novel ideas and is well-written. These are:
- novel approach for training with spectral inputs + spectral evaluation of the Loss via Parseval's identity
- relevant baselines and ablations that show the effect of the individual components
- good motivation for the overall approach

**Weaknesses:**

Some interesting claims are made, for which I would hope for more clarification and theory if possible. These are:
- Why is training in spectral space better? analytically both approaches should be identitcal (due to the identity) and gradients should be the same. Some theoretical analysis on this would have made the paper better.
- how aliasing effects are avoided by using spectral coefficients as input data rather than the signal itself. Ideally, this should come with an ablation study on its own
- the baselines could be better explained - I would like to see parameter counts and spectral resolutions of all approaches.

**Questions:**

Major:
1. NSM is trained using Parseval's identity using a spectral loss. From the results I can see that NSM outperforms CNO (the same but PINN loss). Why is that? Mathematically speaking, the two loss functions are identitcal and so should be the gradients. The only difference I can spot is potentially an added spectral transform in the gradients due to the summation in spectral space. Numerically, this would make a difference but I have a hard time understanding why this should lead to better performance. Any ideas?
2. Results specify resolutions for FNO but do not disclose the spectral resolution/param count for the NSM. I encourage the authors to add this information
3. Figure 2b - why is there only a single point for all resolutions of NSM?

Minor/Clarity:
1. in eq. 5 what is phi? It doesn't seem to be defined
2. The statement that the neural operator is a mapping between grid points is limiting. It is simply realised as such and trained at a fixed resolution but conceptually, nothing stops the user from evaluating it on another grid. I believe this is what the authors mean with "implemented" in the last sentence of page 4 but I find this sentence to be somewhat misleading. A clarification would help in my opinion.

---

> ### Author Response · Authors · 2023-11-16
>
> Thank you for your comments, and we are glad to see that you found our work novel, and appreciated the effectiveness of our approach. We answer your questions below, and have also updated the manuscript in blue to reflect these changes, including the addition of theoretical analysis.
>
> **Question: Why is training in spectral space better? Spectral loss seems to be identical to the grid-based PINN loss. Theoretical analysis?**
>
> The primary reason training in spectral space is better is because of the error from discretization in grid-based methods. The spectral loss has the advantage of computing exact residuals. As the spacing between grid points (grid resolution) goes to infinity, a grid-based PINN loss would be identical to a spectral loss. However, in practical settings, the grid resolution is not going to be continuous, and it is very computationally expensive to train ML models with a fine grid resolution (e.g., expensive backpropagation). In these scenarios, the error from finite grid sizes is not negligible, and leads to the accuracy discrepancy between NSM and CNO.
>
> To demonstrate this, we have added theoretical analysis to the paper, deriving an error analysis for the grid-based PINN loss in Appendix B. While a general theory for all PDEs is out of scope for this work, we look at second-order elliptic PDEs (a class of well-studied PDEs). Our analysis shows that as long as the grid spacing is finite, the expected solution error can be non-zero even if the grid-based PINN loss is minimized arbitrarily well.
>
> **Question: How is aliasing error avoided? Need an ablation study**
>
> As discussed in Section 3.2, the activation function is applied on the function value of the collocation points, which only depend on the number of bases and are independent of input and output resolution. In other words, aliasing is avoided because we are not using input grids. We believe that the accuracy comparison between NSM and CNO+PINN, when tested on different resolutions (see Fig. 2c), already demonstrates this point.
>
> **Question: Parameter counts and spectral resolutions**
>
> We have added parameter counts to Appendix C in the updated paper. For the spectral resolutions, we are using the notation “number of modes”, as it is the conventional notation from FNO. We refer the reviewer to the “Parameters” paragraph in Appendix C for this information.
>
> **Question: Why is there only a single point for NSM in Fig. 2c?**
>
> As shown in Section 3.2, NSM operates exclusively on collocation points, which depends on the number of bases but not on the input resolution. NSM also does not have aliasing error, which guarantees that NSM has the same error regardless of the test resolution (i.e., it has the same error for all input grid sizes).
>
> **Question: What is $\phi$ in Eq. 5?**
>
> Throughout the paper, we use $\phi$ to denote the parameter function in PDE, which is specified in the introduction and the second sentence of Section 3.
>
> **Question: The statement that the neural operator is a mapping between grid points is limiting**
>
> To clarify, we want to emphasize that neural operators are defined and used as mappings between function values on grids, as opposed to our spectral-based ones. Grid-based neural operators can definitely be trained and tested on different resolutions, which is exactly what we show in Section 4.2. We have clarified this in the updated manuscript on page 4.

---

> > ### Comment · Reviewer_4uLV · 2023-11-20
> >
> > I thank the authors for addressing my concerns and clarifying my questions. I have raised my rating to an accept.

---

> > > ### Author Response · Authors · 2023-11-20
> > > **Thank you**
> > >
> > > Thank you for your response!

---

### Official Review · Reviewer_dP1d · 2023-10-27

**Soundness:** 3 good
**Presentation:** 3 good
**Contribution:** 3 good
**Rating:** 8
**Confidence:** 4

**Summary:**

The paper introduces Neural Spectral Methods (NSM) as a novel approach to solving partial differential equations (PDEs). What makes NSM unique in comparison to the numerous ML methods out there is that it's structurally integrated with spectral representations, Parseval's Identity, and orthogonal basis functions, all of which are classic numerical receipes that were well studied. It's a solid step towards hybrid NN and classic numerical method.

**Strengths:**

Like discussed in the summary, the strength of the paper is the elegance integrating NNs with existing spectral methods. In particular, spectral basis are used to represent the functions and NNs are only used to map from spectral coefficients to spectral coefficients (with neural operator architecture). In addition, by leveraging the Parseval's Identity, the proposed method is able to train without using the expensive PDE residual norm (PINN loss). The authors mathematically proved the equivalence of the spectral loss and the PINN loss. This approach avoids sampling a large number of points and the numerical quadrature.

The author backed up these theoretical advantages with experiments that demonstrate 100X training time speedup.

The intro is well written. The short summary on of the the limitations of existing methods (data availability, optimization, computation cost) are very good. And the paragraph above it clearly place the author's work in the literature.

**Weaknesses:**

The paper briefly mentions some limitations of existing ML methods for solving PDEs, but it could benefit from a more extensive discussion of the potential drawbacks and challenges specific to NSM.

Basis are predefined manually. It's unclear how to choose these basis functions.

**Questions:**

Since the proposed resesarch is embedded in the spectral method, it's clearly limited by the issues of many spectral method. For example, as discussed in "Implicit Neural Spatial Representations for Time-dependent PDEs (ICML 2023), spectral methods also have a global support, just like neural networks. So I would be curious to understand more of these limitations. What are some disadvantages of spectral method, in comparison say grid-based neural methods. Also how does the spectral approach compare to Implicit Neural Spatial Representations for Time-dependent PDEs (ICML 2023) which avoids grid completely and only user a MLP architecture.

"Focus on scenarios with no solution data." This is understandable but the scope of the paper. Since the author's method is a very general spectral formulation, why not try it on some data? My biggest concern right now is that the examples shown are all 2D toy example. Would it make sense to try the method on large scale 3D problems where data is already available?

"Dresdner et al. (2022); Xia et al. (2023); Meuris et al. (2023) use spectral representations in the spatial domain. These studies differ in problem settings and deviate from our method in both architecture and training" I looked in details of these papers and can confirm that the author's work is unique. But I think for a complete paper, the author should discuss their similarities and differences.

---

> ### Author Response · Authors · 2023-11-16
>
> Thank you for your insightful review and positive words, and we are glad to see that you found the work elegant. We reply to your comments below, and we have also updated the manuscript (updates in blue),  including an additional experiment on the difficult problem of long temporal transient flow.
>
> **Question: Discussion on challenges and drawbacks specific to NSM**
>
> We have included an additional section in Appendix D, for discussions on NSM challenges and future directions.
>
> **Question: How to choose basis functions**
>
> Similar to traditional spectral methods, the basis functions are chosen in accordance to the boundary conditions in each dimension. In this work, we use Fourier basis on periodic dimensions and Chebyshev polynomials otherwise. In the updated manuscript, we added detailed explanations in Appendix C.1, C.2, and C.3.
>
> **Question: Spectral methods limitations, how does it compare to [1]**
>
> We outline some spectral methods limitations in Appendix D. Regarding how our approach compares to Ref. [1], there are a number of differences:
> 1.  Different problem setting: [1] is in the PINN setting (solving one PDE instance at a time), and therefore is much slower (>10h for training).
>
> 2. Their method is specific to time-dependent problems, whereas our method can be applied to both time-dependent and time-independent problems.
>
> 3. They are not grid-based, but still rely on sampling-based techniques in the spatial domain.
>
> **Question: Focus on scenarios with no solution data**
>
> Our focus on scenarios without solution data is deliberate, stemming from both prior observations and realistic considerations:
>
> 1. **Data-constrained settings are more realistic for real-world applications.** It is more common that one has knowledge of the governing PDEs (or conservation laws, more generally), but lacks corresponding solution data. In many situations, it is very computationally expensive to setup dedicated and expensive numerical solvers and generate new data (as well as gather observational/experimental data).
>
> 2. **Gap in data-driven settings and data-constrained settings.** When current neural operators, such as FNO, are compared in the data-driven setting vs. trained with only physical laws, there is often a large gap between the performance of the two. While it is true that our approach can be applied to the former setting, we leave it to future work to explore this problem in a data-driven setting, especially because it is also rare to have both knowledge of the governing PDEs *and* solution data. We focus on the data-constrained setting because it is both more realistic and because the potential in this setting is far from being fully exploited.
>
> As an additional goal of our work was to demonstrate the effectiveness of our introduced methods like the spectral loss, which is easier to isolate in the data-constrained setting, we also leave adding the data loss to our formulation as future work.
>
> **Question: The examples in the manuscript seem to be 2D toy examples**
>
> We want to first highlight the non-trivialness of our Navier-Stokes experiment, and also that this is technically a 3D setting as there are 2 spatial dimensions and one temporal dimension. This is one of the most challenging fluid dynamics problems, and current state-of-the art ML methods still struggle with this system [2] [3] [4], even in a data-driven setting.
>
> Nevertheless, we have added an additional experiment to further demonstrate the effectiveness of NSM. We look at the long temporal transient flow problem (a 3D problem with 2 spatial dimensions and 1 temporal dimensions). This problem has a long time interval and sharp features, making it extremely difficult to solve with ML models (as well as numerical solvers). We have added this experiment to Section 4.3 in the paper, and demonstrate that NSM has significant accuracy and efficiency improvements compared to FNO + PINN loss (FNO + PINN loss is >50% error, while NSM is <15% error).
>
> **Question: Authors' work is unique but more details should be added to discuss related works (Dresdner et al. (2022); Xia et al. (2023); Meuris et al. (2023))**
>
> We are glad to see that you recognize that our work is unique. We have added more details on the similarities and differences in the updated manuscript in Section 2 (related works).
>
> ---
>
> [1] Implicit neural spatial representations for time-dependent pdes. Chen, Honglin, et al. ICML 2023.
>
> [2] Machine learning–accelerated computational fluid dynamics. Kochkov, Dmitrii, et al.  PNAS 2021.
>
> [3] Physics-informed neural operator for learning partial differential equations. Li et al. arXiv:2111.03794.
>
> [4] Learned coarse models for efficient turbulence simulation. Stachenfeld, Kimberly, et al.  arXiv:2112.15275.

---

> > ### Comment · Reviewer_dP1d · 2023-11-21
> >
> > Thank you for the detailed response! I have raised my score to accept.

---

> ### Author Response · Authors · 2023-11-21
> **Thank you**
>
> Thank you for your response!

---

### Official Review · Reviewer_9377 · 2023-10-29

**Soundness:** 3 good
**Presentation:** 3 good
**Contribution:** 3 good
**Rating:** 8
**Confidence:** 4

**Summary:**

The paper proposed a novel spectral-based neural operator, Neural Spectral Method (NSM), trained by self-supervised loss, i.e., it needs no data. Previous methods including FNO usually are trained with data, though they can also be trained in self-supervised way with PINN loss. This work extends the idea of self-supervised neural operators and proposes a new loss by Parseval's identity.

For periodic and non-periodic boundary conditions, NSM uses Fourier basis and Chebyshev basis respectively.

Under self-supervised setting, the paper compared NSM with some baseline modes, e.g., SNO, FNO combining with PINN loss. It shows that the proposed method has advantages including faster training convergence, higher accuracy, especially at super-resolution inference.

**Strengths:**

**Originality:** The paper has several novelties, including a new design of neural operator with fixed bases. A novel residual loss by Parseval's identity.

**Quality:** The paper has carefully benchmarked the proposed method in training cost, inference cost and accuracy. It exhibits advantages over several baseline methods.

**Clarity:** The paper is well organized and clearly presented.

**Significance:** The proposed method can in inspiring to the community in both

**Weaknesses:**

More comparison can be done, for example, Transform Once

>Poli, Michael, et al. "Transform once: Efficient operator learning in frequency domain." Advances in Neural Information Processing Systems 35 (2022): 7947-7959.

which can be combined with PINN loss or the loss proposed here.

**Questions:**

Regarding aliasing error, it seems to me the solution proposed in the paper is to fix the resolution of grids and do interpolation in higher resolution. If this is correct, it is essentially advising people not to use super resolution inference like FNO does. Hence, this is not an advantage of NSM, but an insight about super resolution, which can be also applied to FNO, i.e., fixing resolution grids + interpolation.

---

> ### Author Response · Authors · 2023-11-16
>
> Thank you for your review. We are glad to see that you found the paper inspiring and had several novelties. We also appreciated that you noticed the strong performance of our method. We have added an additional comparison to T1, and also added an ablation and reply to your other question about fixing grids and interpolation below. We also updated the paper with these changes, which are in blue.
>
> **Question: More comparisons can be done, such as TransformOnce**
>
> We have added an experiment with TransformOnce (T1) as an additional baseline for the Reaction-Diffusion problem. We train T1 with both a PINN loss and our spectral loss. We added these results to the paper, which are detailed in Appendix C.2. We highlight two implications from these experiments:
>
> 1. **NSM consistently has much higher accuracy.** NSM is consistently more accurate than T1 + Spectral loss and T1 + PINN loss, and is significantly more accurate for higher diffusion coefficients. We hypothesize that one reason for this is because the practice of applying non-linear activations directly on the spectral coefficients hurts the model’s robustness (SNO also has a similar issue), and shows the advantages of NSM’s design choices for the base architecture.
>
> 2. **In general, T1 achieves higher accuracy when trained with our spectral loss, rather than the PINN loss.** While T1 + PINN loss has an increase in accuracy as the grid size increases, the overall error is larger than T1 + Spectral loss. This result shows that our spectral training strategy is generally more effective than the PINN loss, and can be applied to various different base architectures.
>
> **Question: Solution proposed in paper is to fix resolution of grids and do interpolation in higher resolution**
>
> For dimensions (e.g., spatial dimensions) using Fourier basis, our approach indeed resembles interpolation on a fixed grid. However, this is not true in other scenarios: for example, for Chebyshev basis (and other bases that may arise), the non-uniformity of collocation points means that we do not do a conventional interpolation. In this sense, the aliasing error can be avoided only if the grid is chosen in accordance with the basis functions.
>
> **Question: Findings seem to suggest that this is more an insight about super resolution that can also be applied to training FNOs (fixing resolution and performing inference by interpolation)**
>
> This is an interesting comment, and we investigated this further. As we noted in our previous response, in many scenarios, we do not fix the resolution and do interpolation at higher resolutions. NSM does have the advantage of being able to learn the higher resolution solution. While we do not explicitly always do this approach (fix resolution + interpolation), we add an ablation that trains FNO on a fixed resolution and interpolates at inference time to higher resolutions. The experiment setting and results are detailed in Appendix C.2.1. As we see, FNO has much higher error with the fixed-grid training strategy. This implies that the spectral training strategy has advantages that are unique to its design, and not easy to replicate in other grid-based base architectures.

---

> > ### Comment · Reviewer_9377 · 2023-11-21
> >
> > Thanks for the reply from the authors.
> >
> > 1. I'm glad to see the additional experiments on T1 as a baseline model for reaction-diffusion problem. Thanks for pointing out that the spectral loss can also improve the performance of T1 compared to PINN loss. How about T1+Spectral loss on the other two problems, NS2D and Poisson?
> >
> > 2. Thanks for the answer regarding aliasing error, particularly the material in Appendix C.2.1. However, I'm still confused about the training method of FNO+PINN. In the ablation study of C.2.1, you mentioned FNO+PINN being trained on fixed grid 32 x 64. Do you intend to compare with non-fixed-grid training? Yet in the beginning of section 4, above equation (13), PINN loss is said to be trained on uniform grids. Please clarify this.
> >
> > Overall, I enjoyed reading this paper and appreciated the work done by the authors, which may be an important work in the community of AI4PDE. The methodology of SSL for neural operators is highly valuable considering the data cost for PDE is usually much higher than many other fields. If you can answer my additional questions, I will raise my score.

---

> ### Author Response · Authors · 2023-11-22
>
> Thank you for your response, and we’re glad that you enjoyed reading our paper! We reply to your two points below:
>
> **Question: T1 + Spectral on Poisson and 2D NS**
>
> We ran these experiments, and have updated the paper with the results of these experiments. These are in Table 1 (Poisson, periodic BC), 7 (Poisson, Dirichlet BC), and Table 10 (2D Navier-Stokes). We see a similar trend as the reaction-diffusion problem, where T1+Spectral does better than T1+PINN (in some cases, much better), though still not as well as NSM.
>
> **Question: Clarify training of FNO+PINN**
>
> This is correct that in Appendix C.2.1, we train FNO+PINN on a fixed grid 32x64 setting. We compare it to the spectral training strategy used by NSM. We then do inference on different grid discretizations. At the beginning of Section 4, when we said that the PINN loss is trained on uniform grids, we meant that it was trained on a fixed rectangular grid (e.g., 64 x 64, 128 x 128, 256 x 256), and with a specific discretization. We have updated the paper to clarify this.

---

> > ### Comment · Reviewer_9377 · 2023-11-22
> >
> > Thanks for your reply! I have updated my score.

---

> > > ### Author Response · Authors · 2023-11-23
> > > **Response to reviewer 9377**
> > >
> > > Thank you for your response, and helping us make our paper better!

---

### Official Review · Reviewer_KbDK · 2023-10-31

**Soundness:** 3 good
**Presentation:** 3 good
**Contribution:** 2 fair
**Rating:** 3
**Confidence:** 4

**Summary:**

The paper introduces Neural Spectral Methods (NSM) for learning solutions to Partial Differential Equations (PDEs) in the spectral domain, with a model that parameterizes spectral transformations. The authors introduce a new class of spectral-based neural operators and leverage Parseval's identity to derive a spectral loss that does not require auto-grad or finite-differences to approximate derivatives. Adopting a *data-constrained* setting, they conducted a set of experiments to validate their approach on three different PDEs.
Their method demonstrates significant speedup and accuracy improvements over considered baselines (FNO + PINN loss, SNO + spectral loss , NSM + PINN loss).

**Strengths:**

The paper is well written and the method section is easy to follow. The spectral loss seems like a promising direction for solving PDEs with Neural Networks. Experimentally, the method outperforms all considered baselines in terms of L2 relative error. Reaction-diffusion and  Navier-stokes experiments were done for different values of diffusion and viscosity coefficients, which highlights the robustness of the method. Figures 2 and 3 show that NSM converges faster during training and is also insensitive to the spatial resolution at inference.

**Weaknesses:**

The motivation of the paper is not transparent to me. The authors propose in this paper two novelties for PDE-based neural networks : a spectral loss and a general design for spectral-based neural operators. While I understand that the first is supposed to simplify the training of PINNs, the second one seems to be a new architecture for solving operator learning tasks like FNO or SNO. Therefore, their method is not a PINN-like solver with a new loss, but rather a deep surrogate model that can approximate the PDE solution from an initial condition or forcing term, without data in the domain. This positioning should be stated explicitly.

As a consequence, the selected baselines and chosen setting cannot lead to a fair comparison between the different methods. Neural operators such as FNO or SNO have been proposed to learn mappings between functions that can be accessed through point-wise evaluations. Therefore, they require data for training and should not be trained with a PDE loss only. I think the authors should focus their comparison with classical PINN methods, and show first that their model is capable of solving a single equation for diverse types of PDEs, and then compare it to PINN approaches that target generalization through meta-learning such as [1], [2].

I also do not understand why NSM is claimed to be insensitive to the grid size. Is a grid ever used for NSM ?  The forcing terms or initial conditions seem to be queried at the collocation points. This avoids aliasing problems rather than preventing them.

There is overall a lack of details regarding the implementation of the proposed method. We do not know if the experiments were done with fourier or chebyschev basis, or both. The truncated number of basis functions considered is also not mentioned.

[1] Meta-Auto-Decoder for Solving Parametric Partial Differential Equations. Huang et. al. Neurips 2022.

[2] Learning differentiable solvers for systems with hard constraints. Geoffrey Négiar, Michael W. Mahoney, Aditi S. Krishnapriyan. ICLR 2023.

**Questions:**

The exposed approach has been tested when $\phi$ is either an initial condition or a forcing term in the equation. While this is encouraging, I wonder if the method would remain applicable if we changed the boundary conditions between samples ? Let's say on a circle or a square. Would there be an issue between the change of dimensions between the 1D input parameter function and the 2D output ?

I am also curious to understand the inference of NSM. For a new input function, is it purely operator-based or do you also finetune the model to reduce the PDE loss ? What kind of guarantee do you provide on the PDE solution for a new parameter function ?

The operators $T$ and $T^{-1}$ are not detailed. What is their complexity with respect to the input length ? How do you choose the collocation points ? Does the number depend on the difficulty level of the PDE ?

Could you elaborate on this point ? ```The Fast Fourier Transform (FFT) in the forward pass of FNO escalates to quadratic in batched differentiation```.

What is $\mathcal{H}$ throughout the paper ?

---

> ### Author Response · Authors · 2023-11-16
>
> Thank you for your comments. We’re glad to see that you found the paper well-written, and appreciated the strong empirical results from our method. Based on your comments, your main concerns seem to be around the motivation of this work. We address all of your comments below, and note that the neural operator trained with a PDE loss is a common setting. We also update the paper with new results, which are in blue.
>
> **Question: Motivation of the paper, neural operators have to be trained with data and can’t be trained with a PDE loss only**
>
> This is actually not the case, and neural operators are commonly trained with a PDE loss only, as highlighted in papers such as [3], [6], [7], and [8]. In fact, the neural operator + PDE loss setting is a much more useful and realistic scenario than classical PINN methods, as now we aim to learn multiple different PDEs mapped to multiple different solutions.
>
> On the point of being less useful and realistic, PINN methods are slow because of the longer training time and only solving one PDE at a time. The setting that we look at is also a more difficult setting (neural operators + PDE loss), but we still have strong performance across problems. Finally, note that even though neural operators learn mappings between functions that can be accessed through point-wise evaluations, this does not mean that they require data for training, as self-supervised PDE losses can also be enforced on the domain.
>
> **Question: Focus comparison on PINN setting and comparing to meta-learning such as [1] and [2]**
>
> As we describe above, the neural operator + PDE loss setting is much more realistic. It is also a much harder learning setting. Despite this, we get low error on *many* PDEs in the test set, as opposed to a single one (as in classical PINN settings). Thus, we believe that we have already demonstrated the performance and capabilities of our method on many different PDEs (going beyond just single equation PDE settings).
>
> We also note that the Reaction-Diffusion experiment in Section 4.2 follows the same setting in [5], where PINNs are shown to completely fail on this problem. Here, we show much stronger performance in the neural operator + PDE loss setting.
>
> We also do not quite understand your comment about comparing to meta-learning PINN approaches like [1] [2]. Ref. [1] looks at a completely different problem setting, and is not comparable to our setting. Ref. [2] does no meta-learning, and is also a different problem setting. We believe that these are out of scope, and we have already compared our method to many other neural operator + PDE loss settings.
>
> **Question: Claim about NSM being insensitive to grid size, is a grid ever used**
>
> The reason NSM is insensitive to the input resolution is precisely because the model does not operate on the input grid throughout its layers. The coefficient function is queried at collocation points, transformed to its spectral form. Similarly, the final prediction is also in its spectral form.
>
> **Question: Details around implementation of the proposed method**
>
> We have added a number of additional details about the experiments to the paper, in Appendix C. We also specify if the experiments were done with Fourier or Chebyshev basis, parameter counts, and modes (number of truncated basis functions).
>
> **Question: Changing BCs between samples and dimensions of the input parameter**
>
> This is an interesting question. Technically, the geometry of the domain can be used as the input, and requires no change in both our model and training technique. But this problem is out of scope for this work, as we specifically look at problems with fixed domains.
>
> In both the Reaction-Diffusion equation and the Navier-Stokes equation, the parameter functions are initial conditions, which have one less dimension than the solution. Following a common practice employed in other works, the initial condition is broadcasted over the time interval to serve as the input function.

---

> > ### Comment · Reviewer_KbDK · 2023-11-22
> > **Response**
> >
> > > This is actually not the case, and neural operators are commonly trained with a PDE loss only.
> >
> > I still disagree with this statement. Neural operators were originally designed for learning mappings between functions with access to data. PINO with a PDE loss only is not considered a standard method for learning a neural operator.
> >
> > > In fact, the neural operator + PDE loss setting is a much more useful and realistic scenario than classical PINN methods.
> >
> > This entirely depends on the definition of usefulness. If you want to have some guarantees on your solution or know when it fails, because you actually know the PDE, then the PINNs method can actually be judged to be more useful. On the other hand, PINN does not indeed generalize or amortize optimization between different trainings, or at least in the vanilla PINN framework.
> >
> > > The setting that we look at is also a more difficult setting (neural operators + PDE loss).
> >
> > I agree that it is a particularly difficult setting, and I am therefore worried that the baselines might not be appropriate.
> >
> > > As we describe above, the neural operator + PDE loss setting is much more realistic.
> >
> > Why do you think it is more realistic ?
> >
> > > We also do not quite understand your comment about comparing to meta-learning PINN approaches like [1] [2].
> >
> > I was simply trying to cite more relevant baselines from the literature. Why do you think that [1] is not the same setting ? I can quote from [2] the following sentences: "Our goal is to learn a mapping between a PDE parameter function $\varphi : \mathcal{X} \rightarrow \mathbb{R}$ and the corresponding PDE solution $u(\varphi) : \mathcal{X} \rightarrow \mathbb{R}$," and "We test the performance of our model on three different scientific problems: 2D Darcy Flow, 1D Burgers’ equation  and 1D convection. In each case, the model is trained without access to any solution data in the interior solution domain." It seems to me that this setting is the same as yours.
> >
> > Overall, I thank the authors for their answers. However, I am still not convinced by the set of baselines, so I will keep my score.

---

> ### Author Response · Authors · 2023-11-16
>
> **Question: NSM inference and performance guarantees**
>
> During inference time, the input function is transformed into spectral coefficients. The model directly predicts the spectral form of the solution, i.e., no finetuning is performed.
>
> As in traditional vision and language learning tasks, performance certification of ML models, i.e., its generalization guarantee, is still an active research field and out of scope of this work. In the context of PDEs, the stability of the equation is the most relevant topic to this question. When the PDE residuals are minimized, the expected solution error can be bounded by the stability constants. We refer the reviewer to [4] as an example of research in this topic.
>
> **Question: Operators T and T^-1**
>
> As discussed in Section 3.2, T is the interpolation operator and T^-1 is the evaluation operator. For Fourier basis and Chebyshev polynomials considered in this work, both bases have sublinear algorithms for T and T^-1. The collocation points are chosen in accordance with the basis functions in each dimension. The number of basis functions is a hyperparameter, which depends on the prior knowledge about the regularity of the solutions.
>
> **Question: Elaborate on “The FFT in the forward pass of FNO escalates to quadratic in batched differentiation.”**
>
> We have added discussion on the complexity of grid-based methods to Appendix A.3. We refer the reviewer to Section 3.3 in [3] for discussions in its original context.
>
> **Question: What is $H$?**
>
> $H^k$ is k’th order Sobolev space. The superscript is omitted for $k=0$, i.e., $H$ is the standard $L^2$ space. We have updated the paper to clarify this notation (Section 3.1).
>
> ---
>
> [1] Meta-Auto-Decoder for Solving Parametric Partial Differential Equations. Huang et. al. NeurIPS 2022.
>
> [2] Learning differentiable solvers for systems with hard constraints. Geoffrey Négiar, Michael W. Mahoney, Aditi S. Krishnapriyan. ICLR 2023.
>
> [3] Physics-informed neural operator for learning partial differential equations. Li et al. arXiv:2111.03794.
>
> [4] Is $L^2$ Physics Informed Loss Always Suitable for Training Physics Informed Neural Network?. Wang, Chuwei, et al. Neurips 2022.
>
> [5] Characterizing possible failure modes in physics-informed neural networks. Krishnapriyan, Aditi, et al. NeurIPS 2021.
>
> [6] Applications of physics informed neural operators. Rosofsky, Shawn G., Hani Al Majed, and E. A. Huerta. Machine Learning: Science and Technology.
>
> [7] PINO-MBD: Physics-Informed Neural Operator for Solving Coupled ODEs in Multi-Body Dynamics. Ding, Wenhao, et al. arXiv:2205.12262.
>
> [8] A novel physics-informed neural operator for thermochemical curing analysis of carbon-fibre-reinforced thermosetting composites. Meng, Qinglu, et al. Composite Structures 2023.

---

> ### Author Response · Authors · 2023-11-23
> **Additional response to reviewer KbDK**
>
> > Why is the neural operator + PDE loss setting more realistic?
>
> As we have discussed, it is more realistic because it is very expensive to generate new data. Our method can work without data, increasing the flexibility when learning new PDEs. In the data-driven setting, solution data needs to be generated numerically or experimentally, which can be expensive or even infeasible. Compared to just the PINNs setting, we are also able to solve multiple PDEs at once, making this setting much more competitive with numerical solvers on the efficiency side (as PINNs require re-training for every new PDE).
>
> Indeed, as Reviewer 9377 pointed out, the self-supervised setting for neural operators is highly valuable, because it is expensive to generate new data.
>
> > Neural operators were originally designed for learning mappings between functions with access to data. PINO with a PDE loss only is not considered a standard method for learning a neural operator.
>
> By this logic, all architectures (Transformers, CNNs, etc.) were originally designed in a supervised learning setting. However, as we have now seen, huge gains have been made by looking at the self-supervised setting. Neural operators are, fundamentally, learned mappings between functions, and the architecture is agnostic to the exact loss formulation.
>
> > This entirely depends on the definition of usefulness. If you want to have some guarantees on your solution or know when it fails, because you actually know the PDE, then the PINNs method can actually be judged to be more useful. On the other hand, PINN does not indeed generalize or amortize optimization between different trainings, or at least in the vanilla PINN framework.
>
> We are again not sure what you mean about your point on usefulness, because we also know the PDE in the neural operator + PDE loss setting. Regarding the solution guarantees, PINNs show no advantage over the operator learning setting. Given a PDE instance, its residuals can be verified on both a trained PINN and neural operator, which give error bounds under certain conditions. Our setting tests for generalization and provides the same guarantees on the minimization of the PDE residuals.
>
> Additionally, as you noted, PINNs don't solve for multiple PDEs, and you have to retrain the model for each new PDE. Thus, the neural operator setting of NSM is strictly more useful from a generalization standpoint.
>
> > The baselines might not be appropriate in this particularly difficult problem setting
>
> We don’t see why the baselines are not appropriate, or why the setting being difficult leads to this conclusion. We compared NSM with a wide range of different neural operators + different PDE loss baselines. For example, PINO [3] is a very standard baseline in the relevant literature, which established promising results combining FNO + PINN loss, and which is an approach that we compare to. During the rebuttal period, we also added the TransformOnce [9], a recently introduced strong baseline model, with both types of loss to demonstrate the effectiveness of our spectral loss.
>
> As we demonstrated, despite the difficult setting, we achieve high accuracy and efficiency with our method. Additionally, we demonstrated that in the reaction-diffusion problem setting, for many grid sizes, we are both more efficient and more accurate than a numerical solver.
>
> > Different problem settings in [1] and [2]
>
> [1] follows the meta learning paradigm, where the PDE parameters are not used as inputs. Instead, for each new PDE instance, the model requires extra fine-tuning iterations to converge. Therefore, these two works are not comparable in terms of computation cost and learning paradigm.
>
> Regarding your reference to [2] as a meta-learning paper, we would like to clarify that it does not fall under the category of meta-learning. Additionally, [2] looks at a hard constraint approach, and as they note themselves, it is hard to scale to more complex problems (particularly non-linear 3D problems, like the ones that we look at here). While a comparison to [2] might be interesting, it is infeasible given that extending [2] to a complex system like Navier-Stokes is a non-trivial research question in and of itself ([2] in its current form would not be able to work on problems like Navier-Stokes because of the scaling issue).
>
> Despite the limited discussion time, we look forward to any further discussions.
>
> ---
>
> [9] Transform once: Efficient operator learning in frequency domain. Poli, Michael, et al. NeurIPS 2022.

---

### Author Response · Authors · 2023-11-16

Dear reviewers,

Thank you for your valuable feedback on our work. We have three major updates to our paper (updates are shown in blue, and modifications in red) that address the questions that we received. We have also added other updates, and we reply individually to each reviewer to describe these.

**We have added the long temporal transient flow as a much harder experiment.** This problem is extremely challenging, as it requires propagating the initial condition over an extended time interval. Our results show that FNO + PINN loss fails with more than 50% error, as reported in the original paper [1]. In contrast, NSM achieves significantly higher accuracy (<15% error), while maintaining a 10x speedup. We refer reviewers to Section 4.3 and Appendix C.3 for experimental details.

**We have included TransformOnce (T1) [2] as an additional baseline architecture.** By training the T1 architecture with both a PINN loss and our spectral loss, we show that although T1 was shown to be more capable than FNO in data-driven settings, NSM consistently outperforms T1 and is significantly more robust. We refer reviewers to Appendix C.2 for experimental details and discussions on the implications of our results.

**We have provided a theoretical error analysis for the grid-based PINN loss.**
Our analysis shows the advantages of using our spectral loss (vs. the grid-based PINN loss). We provide an analysis for second order elliptic PDEs, and show that even if the grid-based PINN loss is minimized, the expected solution error can deviate from zero. We hope this analysis makes our work more complete, by looking at the drawbacks of grid-based methods from a theoretical perspective. We refer reviewers to Appendix B for our formal results and technical proofs.

**Details of each model and experiment.**
Additionally, we have updated the manuscript with implementation clarifications and parameter counts of each model.

We hope these results address your concerns.

---

[1] Physics-informed neural operator for learning partial differential equations. Li et al. arXiv:2111.03794.

[2] Transform once: Efficient operator learning in frequency domain. Poli, Michael, et al. NeurIPS 2022.

---

### Meta-Review · Area_Chair_bcBz · 2023-12-07

**Metareview:**

The paper introduces a novel approach to learning the solution of differential equations in the spectral domain. A spectral basis is employed to represent the underlying functions, and neural networks are trained to learn the mappings between spectral coefficients through a neural operator model. The Parseval identity is utilized to derive a spectral loss, based solely on the optimization of a PDE loss, deviating from the typical data-based approaches of neural operators. The proposed method enables training within a simplified setting compared to objectives such as Physics-Informed Neural Networks (PINNs). Experimental results demonstrate improved accuracy and a significant speedup compared to current Neural Operator (NO) baselines.

In response to the rebuttal, the authors have clarified their experimental setup, introduced a new theoretical analysis, and included additional experiments and baselines. Reviewers acknowledge the novelty and significance of the contribution, considering it a substantial advancement in the field of neural operators. All reviewers, except one, have elevated their scores to an accept.
I recommend accepting the paper.

**Justification For Why Not Higher Score:**

This is a nice paper, with some remaining minor concerns.

**Justification For Why Not Lower Score:**

a

---

### Decision · Program_Chairs · 2024-01-16

Accept (poster)